# A Causal Perspective on Soft Jump-Diffusion for Time-Series Anomaly Detection

## Abstract

Time series anomaly detection is essential for maintaining robustness in dynamic real-world systems. However, most existing methods rely on static distribution assumptions, while overlooking the latent regime-dependent mechanisms and structural shifts that underlie real-world temporal dynamics. This often leads to poor explanation of anomalies and misclassification of environment-induced variations. To address these shortcomings, we propose Causal Soft Jump Diffusion Anomaly Detection (CSJD-AD), a novel framework that models both latent dynamics and soft-gated expected jumps through a structural jump diffusion process. We adopt a causal perspective grounded in environment-conditioned invariance by inferring discrete environment states and condition both the dynamics and jump intensity on them, so the model learns which changes are expected under each regime. By generating paired "expected" (counterfactual) and "observed" (factual) trajectories, the model explicitly contrasts causally consistent behavior with unexplained deviations. Our method achieves state-of-the-art performance across benchmark datasets, demonstrating the importance of incorporating model-implied counterfactual reasoning and jump-aware dynamics into time series anomaly detection.

## 1 Introduction

Time series anomaly detection (TSAD) plays a pivotal role in modern data analysis by identifying unexpected or irregular patterns within sequential data streams. In industry, it enables predictive maintenance by spotting abnormal sensor readings, while in finance, it helps detect fraud through unusual trading behaviors Yang et al. (2024); Livernoche et al. (2023). Beyond these domains, anomaly detection is also indispensable in applications such as quality control, e-commerce analytics, environmental monitoring of smart grids, and Internet of Things infrastructure Pinaya et al. (2022); Yang et al. (2024). As time series data grows in volume and complexity, robust and adaptive detection methods become indispensable. Machine learning and deep models have demonstrated enhanced accuracy and scalability over traditional statistical techniques, navigating challenges such as seasonality, noise, and evolving patterns. Thus, developing advanced, robust, and context-aware anomaly detection models is both timely and essential for maintaining reliability and enabling proactive decision-making in real-world systems Blázquez-García et al. (2021).

Recent advances in deep learning have greatly enhanced TSAD by harnessing neural networks' expressive power. Recurrent models Bontemps et al. (2016); Ergen & Kozat (2019) are widely used to capture temporal dependencies and forecast future values, with deviations from predicted trajectories serving as anomaly indicators. Convolutional Neural Networks (CNNs) Ren et al. (2019); Yang et al. (2023)are also employed to extract local temporal patterns and Transformer Song et al. (2018); Yue et al. (2022) have shown strong performance in long-range sequence modeling, enabling better detection in datasets with complex seasonal and contextual dependencies. Generative approaches, including GAN-based detectors Du et al. (2021); Zhou et al. (2019), have been applied to learn the distribution of normal sequences, using discriminator feedback or likelihood-based scoring to detect anomalies.

Despite these success, most methods assume a stationary data-generating process and overlook latent causal structures, even though real-world environments often exhibit distribution shifts driven by discrete changes

in underlying causal mechanisms Carvalho et al. (2023). Unlike images or event logs with curated labels, most operational time series are raw sensor outputs without environment/state annotations. The regime that determines what is normal is latent, piecewise-constant, and changes at unknown times, so identical observations can be benign in one regime and faulty in another. For example, in industrial monitoring, a spike in machine temperature may be expected during active operational load but highly abnormal during scheduled maintenance or idle states, despite having similar marginal statistics. These context-dependent variations are not anomalies by itself, but reflect different underlying environment regimes. Absent an explicit regime model, detectors routinely misclassify benign shifts as anomalies, degrading alert reliability and obscuring why alarms fire. Reconstruction- and density-based methods are especially vulnerable because they score deviations from a stationary reference rather than regime-conditioned normality.

We address this gap with an environment-aware jump–diffusion model. We introduce a discrete environment variable $E$ that encodes which regime currently governs the system. Conditional on $E$, the main trend and noise (drift and diffusion) are kept stable, so that only violations of this regime are treated as anomalies. Unlike traditional jump–diffusion models Merton (1976) that rely on fixed external shocks, our approach lets the probability of a jump depend on the system's current state. We parameterize this state aware likelihood as $p_E = f_\psi(U, E)$, capturing that anomalies become more or less likely depending on the underlying conditions. Instead of sampling a binary jump, we use a soft gate and add the expected jump effect at each macro-step. This design learns context sensitive jump timing from data, keeps activations sparse by suppressing jumps in stable regimes and increasing them under regime driven volatility, and improves anomaly discrimination by separating structural changes from environment induced variation.

Building upon this environment-aware jump diffusion formulation, we introduce a mechanism for modeling the causal invariance objective through dual latent trajectory generation. In our framework, using only the drift and diffusion terms conditioned on the current environment E, we simulate how the system would evolve if no abrupt perturbation occurred. The counterfactual trajectory, constructed using only these components, models how the system evolves under its current environment regime. As such, it already accounts for all expected or structured changes that arise as part of normal transitions across operating conditions. In contrast, the factual trajectory introduces a jump term that is deterministically weighted via a learned gating propensity. These jumps represent infrequent, irregular deviations that cannot be explained by the environment-driven dynamics alone.

By explicitly separating causally consistent transitions from unexplained deviations, our model, CSJD-AD defines a principled training signal: the discrepancy between factual and counterfactual trajectories quantifies structural violations. This causal contrastive loss focuses learning on environment-invariant irregularities, enhancing the model's sensitivity to meaningful anomalies. In short, the main contributions of this papers are summarized as follows:

- We introduce a discrete environment variable E that, under an invariance perspective, conditions the drift, diffusion, and gated expected jump to disentangle environment-consistent changes from regime-inconsistent deviations.

- We propose an environment-conditioned jump diffusion formulation with a learnable soft gating mechanism that jointly models smooth dynamics and abrupt structural transitions, enabling context-aware and interpretable anomaly detection.

- We construct dual latent trajectories, factual and counterfactual, to improve shift–robust anomaly detection via conditional invariance.

- A unified training objective integrates reconstruction fidelity, variational stability, and causal contrast, resulting in a robust and interpretable framework for TSAD.

## 2 Related Work

**Pattern-Deviation Methods:** This family of methods detect anomalies by measuring how much a subsequence deviates from learned global or local patterns. They define normality based on statistical regularities,

neighborhood structures, or clustering, and flag subsequences as anomalous if they exhibit low likelihood, weak pattern similarity, or sparse local density. For example NormA Boniol et al. (2021), which computes anomaly scores based on the weighted distance of time series subsequences to clustered normal patterns, and Series2Graph Boniol & Palpanas (2020), which constructs a transition graph of subsequence patterns and detects anomalies via low-degree and low-weight graph trajectories.

**Forecasting-based Methods:** These methods Ding et al. (2018); Dai & Chen (2022) detect anomalies by learning to predict future points or subsequences from recent observations. A prediction model is trained on normal data, and anomalies are identified when actual observations deviate significantly from their predicted values. For example, AD-LTI Wu et al. (2020) detects anomalies by combining seasonal decomposition with GRU forecasting and introduces a Local Trend Inconsistency score to account for unreliable historical trends. DeepAnt Munir et al. (2018) is a lightweight CNN-based model that detects point and contextual anomalies with minimal training data and tolerates mild data contamination. GTA Chen et al. (2021) uses transformers and graph convolutions to model temporal and inter-sensor dependencies in multivariate time series for semi-supervised anomaly detection.

**Reconstruction-based Methods:** These methods detect anomalies by learning to reconstruct normal time series patterns. Trained on normal subsequences via sliding windows and latent embeddings, they flag anomalies by identifying high reconstruction errors or low reconstruction probabilities during inference, capturing subtle deviations from expected behavior. For example, VAE-GAN Niu et al. (2020) combines variational autoencoding and adversarial learning to detect anomalies using both reconstruction errors and discriminator feedback in a semi-supervised setting. TranAD Tuli et al. (2022) enhances transformer-based anomaly detection with adversarial training to amplify subtle anomalies and uses self-conditioning to improve stability and generalization.

**Diffusion-based Methods:** Recent methods, such as DiffAD Xiao et al. (2023), $D^3R$ Wang et al. (2023) and IGCL Zhao et al. (2025), all adopt a DDPM-style Ho et al. (2020) generative architecture: they gradually corrupt the input with Gaussian noise and train a denoiser to reconstruct clean sequences, treating large reconstruction or density-ratio deviations as anomaly signals. In contrast, CSJD-AD does not use a DDPM denoising chain. We model the latent dynamics with a jump–diffusion SDE, where anomalies are detected as structural violations of the inferred environment-conditioned dynamics, making our approach conceptually and architecturally distinct from DDPM-based TSAD.

## 3 Methodology

### 3.1 Problem setting

We address TSAD under both semi-supervised and unsupervised paradigms. In the semi-supervised setting, the model is trained on normal data to learn a representation of typical temporal dynamics, then identifies deviations caused by faults or external disruptions as anomalies during inference. Formally, given an observed series $X = \{x_1, \ldots, x_T\}$ with $x_t \in \mathbb{R}^d$ ($d = 1$ for univariate and $d > 1$ for multivariate cases), the objective is to capture the structure of normal behavior and detect departures. We also evaluate our approach in an unsupervised setting, where anomalies are detected solely based on the intrinsic properties of the data without reliance on labeled normal samples.

### 3.2 Variational Causal Encoder

Given an observed time series segment $X \in \mathbb{R}^{T \times d}$, our goal is to encode it into two types of latent representations: a latent mapping matrix $U \in \mathbb{R}^{T \times k}$ capturing the temporal data's underlying dynamics, each row $U_t$ summarizes step-t latent features and each column indexes a latent channel shared across the window, and a discrete environment variable $E \in \Delta^{K-1}$ (a probability simplex over $K$ environments) that serves as an unsupervised index for regime-conditioned latent dynamics and conveys causal semantics through conditioning and counterfactual-style simulation rather than structural identification.

We achieve this via a shared neural encoder $\text{CausalEncoder}(X)$ that produces the variational posteriors: $q_\phi(U \mid X), \quad q_\phi(E \mid X)$, where $\phi$ denotes the encoder parameters. Specifically, $U$ is sampled from a Gaussian

distribution with learnable mean and variance:

$$q_\phi(U \mid X) = \mathcal{N}(\mu_U(X), \text{diag}(\sigma_U^2(X))), \tag{1}$$

and $E$ is sampled using the Gumbel-Softmax reparameterization to approximate a categorical distribution in a differentiable manner, yielding a soft one-hot vector that is then used to to condition downstream network components:

$$q_\phi(E \mid X) = \text{GumbelSoftmax}(\text{logits}_E(X)), \tag{2}$$

each regime $k$ therefore parameterizes its own drift, diffusion, and jump functions; we use the subscript $(\cdot)_E$ to denote conditioning on $E$, conveying causal semantics via environment-conditioned invariances.

The shared encoder is not intended to make $U$ and $E$ semantically identical, but to provide a common temporal feature backbone from the same input window. This ensures that both latent quantities are inferred from consistent observational evidence, while avoiding the computational redundancy of maintaining two separate sequence encoders. Their separation is introduced after this shared representation through different posterior heads, output supports, and downstream usage: one branch provides the continuous latent state used for stochastic trajectory evolution, whereas the other provides a compact regime code that selects the environment-conditioned mechanisms. Thus, parameter sharing improves consistency and efficiency without collapsing the distinct roles of state representation and regime selection.

### 3.3 Causal Soft Jump Diffusion SDE

Building on our motivation, the need to separate causally consistent regime shifts from regime-inconsistent deviations, we now present the formal dynamics that drive our latent representations. Throughout, we retain the intuition of jump diffusion processes while adopting fully differentiable formulation.

#### 3.3.1 From Jump Diffusion to Soft Jump Diffusion

In the classical jump diffusion framework Merton (1976), the latent state $U_t \in \mathbb{R}^d$ evolves according to

$$dU_t = \mu(U_t)\,dt + \sigma(U_t)\,dW_t + J(U_t)\,dN_t, \tag{3}$$

where $N_t$ is a Poisson process of fixed rate and each jump contributes a discrete increment of size $J_E(U_t)$. However, the discrete sampling to Poisson variables breaks gradient flow, complicating end-to-end learning. Neural Jump SDEs Jia & Benson (2019) and the NJDTPP Zhang et al. (2024) are built for event prediction, they tie jumps to observed event and train by maximizing event-time likelihood, which limits their use on densely sampled sensor streams without event timestamps.

To reconcile expressive power with differentiability and utilize a mechanism to decide dynamically when a jump should or should not occur based on the current context, we model the latent state $U_t \in \mathbb{R}^d$ as a continuous diffusion process interspersed with instantaneous soft jumps at macro-grid times $\{\tau_j\}_{j=0}^J$. Concretely, for $j = 0, \ldots, J-1$ we write

$$\begin{cases} dU_t = \underbrace{\mu_E(U_t, E)}_{\text{drift net}}\,dt + \underbrace{\sigma_E(U_t, E)}_{\text{diffusion net}}\,dW_t, \\ U_{\tau_{j+1}} = U_{\tau_{j+1}^-} + \underbrace{p_E(U_{\tau_{j+1}^-}, E)}_{\text{soft gate}} \underbrace{J_E(U_{\tau_{j+1}^-}, E)}_{\text{jump net}}, \end{cases} \tag{4}$$

where $U_{\tau_{j+1}^-} = \lim_{t \uparrow \tau_{j+1}} U_t$.

For each $t \in (\tau_j, \tau_{j+1}]$, the latent state follows a diffusion driven by a Brownian motion $W_t$. At $t = \tau_{j+1}$, we apply an instantaneous soft jump of magnitude $p_E(U_{\tau_{j+1}^-}, E)\,J_E(U_{\tau_{j+1}^-}, E)$. In practice, we restrict to one expected soft jump per window. This aligns the model with the windowing granularity used for evaluation and serves as a first-moment approximation of the cumulative jump effect of a compound-Poisson process over the window, $\int J\,dN \approx p_E J_E$. Here the scalar gate $p_E \in (0, 1)$ encodes the propensity of a structural

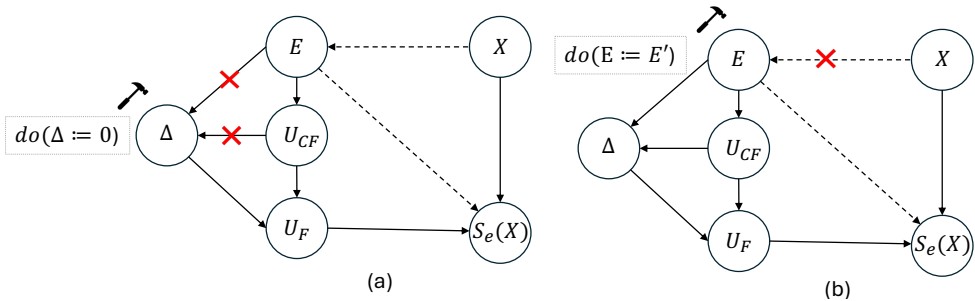

Figure 1: Operational intervention semantics of CSJD on the model computation graph. $do(\cdot)$ denotes clamping an internal node in the computation graph (incoming edges removed). (a) Jump suppression via $do(\Delta := 0)$ isolates diffusion-only dynamics under a fixed inferred regime. (b) Regime override via $do(E := E')$ clamps the mechanism selector and recomputes the induced score while keeping the observed window fixed.

shock conditioned on the current latent state and environment $E_{t_k}$, whereas $J_E$ specifies its direction and scale. Positive entries are excitatory and negative entries are inhibitory. Detailed statements and proofs are provided in Appendix B.

Besides, the use of a single soft jump per window does not restrict the number of underlying jump events. Over one macro-step, the compound-Poisson jump term aggregates all shocks occurring within the window; we approximate this cumulative contribution by its conditional expectation $p_E J_E(U, E)$. Thus, multiple physical shocks inside a window are represented by a single effective jump in latent space that captures their net first-moment impact. A formal justification of this approximation is provided in Appendix B.4.

### 3.4 Causal Path Generation

We evolve the window-level matrix state $U_s \in \mathbb{R}^{T \times k}$ along solver time $s \in [0, 1]$ while preserving its $T \times k$ layout. Given the encoded latent state $U_0$ and environment $E$, we evolve the process over $M$ micro-steps of size $\delta t = \Delta t / M$:

$$U^{(m+1)} = U^{(m)} + \mu_E\big(U^{(m)}, E\big)\, \delta t + \sigma_E\big(U^{(m)}, E\big)\sqrt{\delta t}\, \epsilon_m,$$
$$\epsilon_m \sim \mathcal{N}(0, I), \tag{5}$$

with $U^{(0)} = U_0$. The last state $U^{(M)}$ is the no-jump counterfactual trajectory latent, denoted $U_{\text{CF}}$.

Then we inject the expected jump contribution,

$$U_{\text{F}} = U_{\text{CF}} + p_E\big(U_{\text{CF}}, E\big)\, J_E\big(U_{\text{CF}}, E\big), \tag{6}$$

where $p_E \in (0, 1)$ is a learnable propensity and $J_E$ encodes jump magnitude.

Finally, we blend the two trajectories $U_{\text{final}} = U_{\text{CF}} + \gamma(U_{\text{F}} - U_{\text{CF}})$, $\gamma \in [0, 1]$, where $U_{\text{F}} - U_{\text{CF}} = p_E J_E$, and reconstruct the observation via $X_{\text{gen}} = \text{Decoder}(U_{\text{final}})$, where $\gamma$ controls the strength of jump influence, allowing smooth interpolation between counterfactual and factual paths. We define $\gamma$ as a hyperparameter and the settings are provided in Table 8 of Appendix D. This soft-jump formulation preserves the causal intuition of jump diffusion, $p_E J_E$ increases only in volatile regimes while remaining fully differentiable.

### 3.5 Operational causal semantics and model-implied interventions

We formalize Figure 1 via an intervention semantics on the model's computation graph. Here "causal" refers to counterfactual behavior induced by clamping internal variables (mechanism selection and jump injection), rather than identifiability of real-world causal effects from observational time series. In Figure 1, solid edges represent forward mechanism flow inside the learned model, while dashed edges indicate inference/conditioning (e.g., $X \to E$ and code-conditioned scoring).

Given a window $X$, the encoder defines a posterior over a discrete regime code $E \sim q_\phi(E \mid X)$ and an initial latent state (or its distribution) denoted by $U_0$. For any chosen code $e$, the regime-conditioned drift–diffusion yields a diffusion-only endpoint $U_{CF}(e) = \mathsf{DD}_e(U_0; \theta)$, and the jump module outputs an expected contribution $\Delta(e) = p_e\big(U_{CF}(e)\big) \cdot J_e\big(U_{CF}(e)\big)$, giving $U_F(e) = U_{CF}(e) + \Delta(e)$. Our anomaly score is the window energy $S(X) = \mathcal{L}_{\mathrm{total}}(X)$. To state interventions precisely, we define the clamped-code score family

$$S_e(X) = \mathcal{L}_{\mathrm{total}}\big(X; \text{ all regime-conditioned modules clamped to } e\big), \tag{7}$$

where the observed input window $X$ (and thus the encoder outputs such as $U_0$) is kept fixed and only downstream mechanisms are recomputed under clamping. The default score uses the inferred code ($\hat{e}(X) = \arg\max_e q_\phi(e \mid X)$), hence $S(X) = S_{\hat{e}(X)}(X)$. This yields a minimal falsifiable implication: if the code were redundant, then $S_e(X)$ would be approximately invariant to $e$ for typical $X$; systematic sensitivity of $S_e(X)$ to code overrides indicates that $E$ acts as a non-trivial mechanism selector.

With these definitions, Figure 1 corresponds to two computation-graph surgeries. (i) No-jump counterfactual: $\mathrm{do}(\Delta := 0)$ disables jump injection under the same inferred code $e = \hat{e}(X)$, yielding $U_F^{(0)}(e) = U_{CF}(e)$ and the corresponding counterfactual energy $S_e^{(0)}(X)$. This provides an operational baseline that isolates jump-induced deviations, aligning with our contrastive objective. (ii) Regime-code override: $\mathrm{do}(E := E')$ replaces $\hat{e}(X)$ by a prescribed code $e'$ and recomputes $S_{e'}(X)$. In Section 4.3.2, we instantiate $E'$ using single-$E$ (a constant code for all windows) and Shuffled-$E$ (a permutation of inferred codes across windows). The consistent degradation under these overrides serves as a direct falsification test of redundancy: clamping the code changes the induced conditional dynamics and energy landscape, demonstrating that $E$ is used non-trivially at inference time rather than acting as an auxiliary label. We use $\mathrm{do}(\cdot)$ purely to denote clamping internal nodes (optionally under a fixed noise realization when stochasticity is present), not interventions on the data-generating process.

### 3.6 Inference Training with Counterfactual Loss

Building on the dual-path simulation framework, we introduce a principled loss formulation that enforces meaningful causal representations and stable variational inference.

**Causal Discrepancy Weight.** Before introducing the losses, we define a causal discrepancy weight that modulates the contrastive term during training. It quantifies the magnitude of the environment-conditioned jump and its improbability. The weight is

$$\mathcal{W}_{\mathrm{CD}}(X) = \|J_E\|_1 \cdot (1 - p_E). \tag{8}$$

**Loss Components.** We minimize the total loss:

$$\mathcal{L}_{\mathrm{total}} = \mathcal{L}_{\mathrm{recon}} + \lambda_{\mathrm{causal}} \cdot \mathcal{L}_{\mathrm{causal}} + \lambda_{\mathrm{kl}} \cdot (\mathcal{L}_{\mathrm{KL}}^U + \mathcal{L}_{\mathrm{unif}}^E), \tag{9}$$

where $\lambda_{\mathrm{causal}}$ and $\lambda_{\mathrm{kl}}$ control regularization strength.

The reconstruction loss $\mathcal{L}_{\mathrm{recon}} = \|X - X_{\mathrm{gen}}\|_2^2$, ensures that latent variables capture observable patterns, enabling anomaly detection via reconstruction error.

The causal contrastive loss $\mathcal{L}_{\mathrm{causal}} = \|U_F - U_{CF}\|^2 \cdot \mathcal{W}_{\mathrm{CD}}(X)$, promotes consistency between factual and counterfactual latent paths in stable regimes, while allowing divergence when jumps occur, regularizing behavior and supporting unsupervised anomaly scoring.

Finally, the KL terms Kingma & Welling (2013); Pereyra et al. (2017) regularize the latent space and the environment variable:

$$\mathcal{L}_{\mathrm{KL}}^U := D_{\mathrm{KL}}\big(q_\phi(U \mid X) \,\big\|\, \mathcal{N}(0, I)\big),$$
$$\mathcal{L}_{\mathrm{unif}}^E := \mathbb{E}_{q_\phi(E \mid X)}\left[\sum_{e=1}^{K} E_e \log\big(E_e + \varepsilon\big)\right], \tag{10}$$

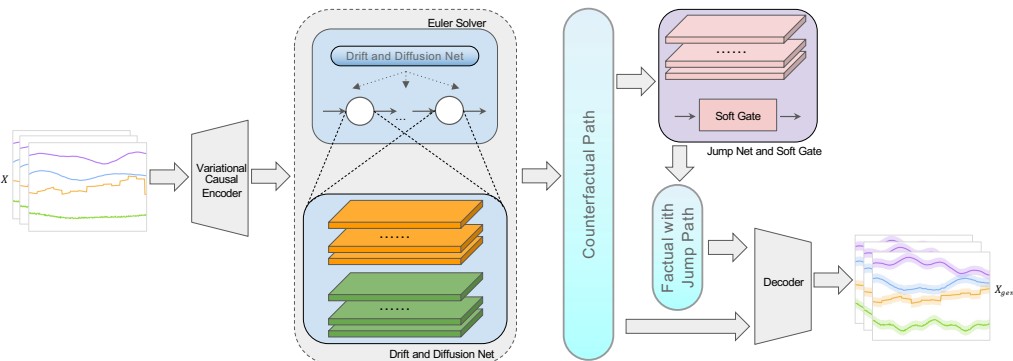

Figure 2: Overall model structure. $X$ is encoded into latent $U$ and environment $E$, evolved by drift–diffusion dynamics to a counterfactual path, perturbed by a gated jump to yield the factual path, blended into $U_{\text{final}}$, and finally decoded back to $X_{\text{gen}}$.

Table 1: Statistics of the ten anomaly-detection datasets. AR stands for Anomaly Ratio.

| Dataset | Dimension | Entity | Train | Test | AR |
|---------|-----------|--------|-------|------|-----|
| ASD | 19 | 12 | 37,089 | 49,452 | 0.295 |
| ECG | 2 | 9 | 6,999 | 2,851 | 0.153 |
| MSL | 55 | 27 | 58,317 | 73,729 | 0.198 |
| SMAP | 25 | 55 | 140,825 | 444,035 | 0.131 |
| SMD | 38 | 28 | 878,560 | 702,848 | 0.042 |
| SWaT | 51 | 1 | 496,800 | 449,919 | 0.123 |
| WADI | 127 | 1 | 784,173 | 172,604 | 0.073 |
| PSM | 25 | 1 | 132,481 | 87,841 | 0.278 |
| Yahoo | 1 | 56 | 30,456 | 7,614 | 0.036 |
| KPI | 1 | 26 | 396,211 | 566,316 | 0.095 |

here $\mathcal{L}_{\text{unif}}^{E}$ is a negative-entropy regularizer on $q_\phi(E \mid X)$, promoting smooth latent representations. Together, these losses ensure stable training and improve generalization across diverse regimes.

### 3.7 CSJD-AD Overall Pipeline

Figure 2 shows the full CSJD-AD pipeline. The variational encoder maps each input window $X$ to a latent state U and a environment code E. A drift network $\mu_E(\cdot)$ and diffusion network $\sigma_E(\cdot)$, both conditioned on E, advance U through an Euler–Maruyama step to generate the counterfactual trajectory $U_{\text{CF}}$. A jump module, likewise conditioned on E, outputs a jump magnitude $J_E(U_{\text{CF},E})$ and gate $p_E(U_{\text{CF},E})$; adding the gated jump yields the factual state $U_{\text{F}}$ as described in equation 6. The model blends the two paths via a coefficient $\gamma$ to obtain the final latent $U_{\text{final}}$, which the decoder transforms back into $X_{\text{gen}}$. Training minimizes the reconstruction error, the causal contrastive loss, and KL regularization on both U and E. At inference, we use the total objective as an energy score, $\mathcal{S}(X) = \mathcal{L}_{\text{total}}(X)$, treating the KL-type terms as data-dependent posterior complexity penalties; constants (e.g., $\log K$) are dropped and $\lambda_{\text{causal}}, \lambda_{\text{kl}}$ are fixed from training. For completeness, we report the variant in Table 7 in Appendix D.

### 3.8 Practical Considerations of TSAD Benchmarks

Public TSAD benchmarks impose two structural limitations that make "semantic environments" and automatic regime-count discovery poorly posed. First, many datasets provide limited feature semantics: channel names are anonymized or dataset-specific, engineering context is absent, and the same numerical channel

Table 2: Time-series anomaly-detection performance on ten public benchmarks (higher is better). Best scores are in **bold**; second–best are underlined. The Table 10 in Appendix include Precision and Recall as an extended version.

| Method | Metric | Multivariate Benchmarks | | | | | | | | Univariate Benchmarks | |
|---|---|---|---|---|---|---|---|---|---|---|---|
| | | ASD | ECG | MSL | SMAP | SMD | SWaT | WADI | PSM | Yahoo | KPI |
| LSTM-VAE | F1 | 0.327 | 0.274 | 0.407 | 0.437 | 0.367 | 0.762 | 0.248 | 0.465 | 0.326 | 0.182 |
| | AUCPR | 0.245±.180 | 0.206±.150 | 0.285±.249 | 0.258±.305 | 0.395±.257 | 0.713 | 0.139 | 0.441 | 0.255±.152 | 0.135±.120 |
| OmniAnomaly | F1 | 0.238 | 0.216 | 0.271 | 0.325 | 0.459 | 0.762 | 0.229 | 0.338 | 0.340 | 0.201 |
| | AUCPR | 0.175±.132 | 0.154±.152 | 0.149±.182 | 0.115±.129 | 0.365±.202 | 0.713 | 0.120 | 0.391 | 0.245±.218 | 0.140±.010 |
| AnomalyTran | F1 | 0.425 | 0.464 | 0.344 | 0.407 | 0.304 | 0.737 | 0.102 | 0.403 | 0.372 | 0.303 |
| | AUCPR | 0.281±.201 | 0.306±.221 | 0.236±.237 | 0.264±.315 | 0.273±.232 | 0.681 | 0.040 | 0.471 | 0.261±.182 | 0.204±.139 |
| TranAD | F1 | 0.305 | 0.461 | 0.420 | 0.471 | 0.386 | 0.310 | 0.263 | 0.568 | 0.484 | 0.287 |
| | AUCPR | 0.238±.178 | 0.368±.251 | 0.278±.239 | 0.287±.300 | 0.412±.260 | 0.192 | 0.139 | 0.669 | 0.691±.324 | 0.285±.206 |
| $D^3$R | F1 | 0.253 | 0.301 | 0.197 | 0.217 | 0.326 | 0.232 | 0.117 | 0.410 | 0.201 | 0.152 |
| | AUCPR | 0.150±.110 | 0.180±.131 | 0.138±.101 | 0.445±.218 | 0.228±.167 | 0.205 | 0.070 | 0.507 | 0.120±.080 | 0.090±.061 |
| PUAD | F1 | 0.351 | 0.382 | 0.384 | 0.275 | 0.364 | 0.254 | 0.259 | 0.422 | 0.301 | 0.284 |
| | AUCPR | 0.280±.203 | 0.304±.221 | 0.307±.102 | 0.319±.082 | 0.291±.210 | 0.271 | 0.155 | 0.449 | 0.240±.172 | 0.224±.152 |
| NPSR | F1 | 0.350 | 0.451 | 0.373 | 0.164 | 0.372 | 0.379 | 0.613 | 0.438 | 0.550 | 0.321 |
| | AUCPR | 0.281±.201 | 0.405±.281 | 0.336±.241 | 0.284±.142 | 0.335±.245 | 0.296 | 0.552 | 0.444 | 0.495±.344 | 0.288±.160 |
| DiffAD | F1 | 0.135 | 0.203 | 0.047 | 0.298 | 0.035 | 0.136 | 0.221 | 0.409 | 0.241 | 0.140 |
| | AUCPR | 0.523±.013 | 0.552±.025 | 0.321±.102 | 0.241±.083 | 0.102±.031 | 0.083 | 0.432 | 0.482 | 0.293±.124 | 0.231±.042 |
| Dual-TF | F1 | 0.661 | 0.538 | 0.127 | 0.163 | 0.287 | 0.212 | 0.551 | 0.506 | 0.725 | 0.330 |
| | AUCPR | 0.628±.212 | 0.511±.182 | 0.124±.126 | 0.141±.082 | 0.215±.074 | 0.171 | 0.523 | 0.354 | 0.689±.234 | 0.314±.107 |
| Sensitive-HUE | F1 | 0.366 | 0.309 | 0.451 | 0.251 | 0.397 | **0.904** | 0.699 | 0.381 | 0.281 | 0.170 |
| | AUCPR | 0.340±.188 | 0.410±.245 | 0.432±.121 | 0.319±.194 | 0.462±.283 | **0.873** | 0.641 | 0.681 | 0.489±.429 | 0.227±.253 |
| IGCL | F1 | 0.022 | 0.094 | 0.223 | 0.208 | 0.208 | 0.718 | 0.014 | 0.432 | 0.201 | 0.208 |
| | AUCPR | 0.079±.066 | 0.183±.141 | 0.179±.102 | 0.181±.061 | 0.126±.132 | 0.691 | 0.218 | 0.461 | 0.300±.277 | 0.206±.198 |
| RedLamp | F1 | 0.205 | 0.165 | 0.284 | 0.187 | 0.113 | 0.153 | 0.624 | 0.043 | 0.299 | 0.057 |
| | AUCPR | 0.154±.103 | 0.200±.196 | 0.199±.290 | 0.321±.306 | 0.128±.140 | 0.083 | 0.564 | 0.454 | 0.653±.409 | 0.089±.129 |
| Ours | F1 | **0.676** | **0.584** | **0.529** | **0.478** | **0.575** | 0.763 | **0.716** | **0.686** | **0.966** | **0.346** |
| | AUCPR | **0.682±.193** | **0.631±.176** | **0.464±.296** | **0.400±.317** | **0.637±.183** | 0.783 | **0.644** | **0.774** | **0.938±.200** | **0.342±.242** |

can correspond to different physical quantities across benchmarks. This limits the extent to which any method can claim human-interpretable causal drivers from the raw data alone, and it makes interpretability evaluations inherently non-uniform across datasets. For this reason, we use E as a semantic-free mechanism index, a compact discrete selector that modulates regime-conditioned dynamics, rather than an exogenous variable that must align with a unique physical meaning.

Second, most TSAD benchmarks are multi-entity and lack a globally aligned timeline: entities are recorded with different start times, missing periods, and non-synchronized timestamps, so concatenating entities into a single sequence is generally invalid and can introduce artificial discontinuities. Consequently, regime discovery must operate within each entity and then generalize across entities without a shared clock. We did attempt to automatically infer the regime cardinality and assignments using two representative families of approaches: (i) Bayesian nonparametrics (Dirichlet-process style mixture modeling via truncation) and (ii) explicit model selection over K (training multiple candidates with grid search and selecting by an unsupervised criterion). In our setting, both directions substantially increased computational and engineering cost, requiring multiple training runs and repeated SDE simulations per dataset and per entity, yet did not yield a consistent improvement in anomaly detection performance. Since our primary objective is anomaly detection rather than regime enumeration, we therefore treat K as a small capacity parameter for coarse regime partitioning, fix it to a modest value, and verify stability over a small grid of K values. Finally, the environment-override diagnostics in Table 4 show that the inferred E is used non-trivially at inference time, supporting its role as a mechanism index rather than a redundant auxiliary head.

# 4 Experiment

## 4.1 Experiment Setup

**Benchmark Datasets.** We evaluate our model on eight multivariate time series anomaly detection datasets: ASD Li et al. (2021), ECG Keogh et al. (2005), MSL Hundman et al. (2018), SMAP Hundman et al. (2018), SMD Su et al. (2019), SWaT Ahmed et al. (2017), WADI Ahmed et al. (2017), PSM Abdulaal et al. (2021), and two univariate datasets: Yahoo Laptev et al. (2015) and KPI Li et al. (2022), all with point-wise anomaly labels. Table 1 illustrates the details for each dataset. The multivariate datasets follow a semi-supervised setting, assuming access to anomaly-free training data. In contrast, the univariate datasets lack predefined train/test splits, requiring manual partitioning. As a result, we cannot guarantee the absence of anomalies in the training sets, placing these datasets in an unsupervised setting. For Yahoo and KPI, we exclude entities that contain no anomalies in the test set, since the F1 score would otherwise be undefined.

**Evaluation Metrics.** We evaluate each model using the standard F1 score and the average Area Under the Precision-Recall Curve (AUCPR) across entities. We do not use point-adjusted F1 because it credits an entire anomaly segment when any single point crosses the threshold, which can push random or diffuse predictions to high F1 and inflate scores on long segments Kim et al. (2022). Many datasets contain multiple entities without aligned timestamps, so we train models separately for each. Since F1 is not additive, unlike many baselines that average per-entity F1, we aggregate true/false positives and negatives across entities and recompute the F1 from the combined confusion matrix. For multi-entity datasets, we report the mean and standard deviation of AUCPR. For SWaT, WADI, and PSM, which have only one entity, AUCPR standard deviation is not available. Appendix A.3 (Table 6) reports runtime, memory, and FLOPs comparisons, where CSJD-AD performs competitively.

**Baseline Models.** We evaluate eleven TSAD methods, including VAE-based models (LSTM-VAE Park et al. (2018), OmniAnomaly Su et al. (2019)), transformer-based approaches (TranAD Tuli et al. (2022), PUAD Li et al. (2023), AnomalyTran Lai et al. (2023a), NPSR Lai et al. (2023b), Dual-TF Nam et al. (2024), Sensitive-HUE Feng et al. (2024)), diffusion-based models DiffAD Xiao et al. (2023), $D^3R$ Wang et al. (2023) and IGCL Zhao et al. (2025), and a CNN-MLP model RedLamp Obata et al. (2025).

## 4.2 Overall Experiment Results

Table 2 reports F1 and AUCPR. The extended table with precision and recall, together with training settings and resource usage, appears in Appendices A and D. We fix the window size to 200 on all datasets to limit hyperparameter effects. While many baselines tune the window per dataset, our model delivers strong and consistent results without such tuning.

Our model achieves state-of-the-art performance across all time series anomaly detection benchmarks, consistently outperforming existing methods in both F1 and AUCPR metrics. As shown in Table 2, our model demonstrates strong robustness under severe class imbalance. For instance, on the Yahoo and ECG datasets—both characterized by extremely low anomaly ratios—we achieve AUCPR scores of 0.937 and 0.627, respectively. These represent relative improvements of over 20% compared to the next-best models (TranAD with 0.691 on Yahoo and Dual-TF with 0.511 on ECG), highlighting the model's superior ability to maintain precision and recall in imbalanced settings.

Beyond multivariate benchmarks, our method performs strongly on univariate datasets, demonstrating adaptability across data regimes and flexibility in semi-supervised and unsupervised settings over a broad range of temporal dimensionalities.

## 4.3 Ablation Study

### 4.3.1 Components Effectiveness

We evaluate four ablated variants of our model by disabling each key component in isolation, while keeping all other settings fixed. The w/o $U_{\mathrm{F}}$ variant removes the factual path $U_{\mathrm{F}}$ and omits the causal loss $\mathcal{L}_{\mathrm{causal}}$

Table 3: Ablation study of the proposed model. Each column reports F1 scores (higher is better). **Bold** numbers denote the best result for that dataset. The Table 11 in Appendix is the extended version that includes the AUCPR results.

| Variant | ASD | ECG | MSL | SMAP | SMD | SWaT | WADI | PSM | Yahoo | KPI |
|---|---|---|---|---|---|---|---|---|---|---|
| w/o $U_F$ | 0.562 | 0.573 | 0.523 | 0.443 | 0.563 | 0.751 | 0.691 | 0.672 | 0.892 | 0.194 |
| w/o $p_E$ | 0.675 | **0.604** | 0.515 | 0.432 | 0.568 | 0.742 | 0.702 | 0.651 | 0.899 | 0.310 |
| w/o $E$ | 0.571 | 0.552 | 0.505 | 0.403 | **0.575** | 0.733 | 0.706 | 0.673 | 0.811 | 0.118 |
| w/o $\mathcal{L}_{\text{causal}}$ | **0.682** | 0.506 | 0.507 | 0.451 | 0.563 | 0.759 | 0.711 | 0.662 | 0.932 | 0.250 |
| w/o $\mathcal{L}_{\text{KL}}$ | 0.680 | 0.578 | 0.512 | 0.471 | 0.571 | 0.741 | 0.710 | 0.679 | 0.934 | 0.188 |
| Default | 0.676 | 0.584 | **0.529** | **0.478** | **0.575** | **0.763** | **0.716** | **0.686** | **0.966** | **0.346** |

Table 4: Effect of modifying learned environment settings on F1 performance (higher is better). Best scores for each dataset within a block are shown in **bold**. Table 12 in Appendix is an extended version that includes AURCPR. Default corresponds to training with correctly learned environment variables.

| Strategy | ASD | ECG | MSL | SMAP | SMD | SWaT | WADI | PSM | Yahoo | KPI |
|---|---|---|---|---|---|---|---|---|---|---|
| Single E | 0.654 | 0.581 | 0.512 | 0.450 | **0.575** | 0.759 | 0.709 | 0.663 | **0.966** | 0.306 |
| Shuffled E | 0.539 | 0.467 | 0.504 | 0.408 | 0.396 | 0.732 | 0.698 | 0.651 | 0.811 | 0.143 |
| Default | **0.676** | **0.584** | **0.529** | **0.478** | **0.575** | **0.763** | **0.716** | **0.686** | **0.966** | **0.346** |

accordingly. The w/o $p_E$ variant removes the gating head and injects a deterministic jump once per window. The w/o $E$ variant replaces the causal encoder with a standard encoder which only outputs the continuous latent variable $U$, and disabling the KL loss $\mathcal{L}_{\text{KL}}$. In the w/o $\mathcal{L}_{\text{causal}}$, we retain the computation of $U_F$ and $U_{\text{CF}}$ but omit the trajectory contrastive learning objective during training. Finally, the w/o $\mathcal{L}_{\text{KL}}$ variant disables both the Gaussian-prior KL penalty and the entropy regularization, while preserving the causal encoder that extracts the environmental variable $E$.

As shown in Table 3, most ablations lead to a consistent decline in detection performance across all ten datasets. Notably, some ablated variants still achieve performance comparable to existing state-of-the-art methods on certain benchmarks. For instance, on the SMD dataset, the w/o $E$ variant performs similarly to the full model, suggesting that SMD may contain a single dominant environmental regime, thereby diminishing the benefit of explicit environment modeling. Additionally, on the ASD dataset, removing $\mathcal{L}_{\text{causal}}$ and $\mathcal{L}_{\text{KL}}$ constrain yields slightly better performance; however, the environmental variable $E$, the factual path $U_F$, and the counterfactual path $U_{\text{CF}}$ remain essential components, as their presence continues to support overall model performance.

### 4.3.2 Causal Environment Representation Quality

We test whether the encoder discovers discrete regimes by projecting the learned embeddings with UMAP and clustering with K-Means using the preset $E$. We run K-Means with K equal to the pre-specified environments and color each point by its cluster assignment. As Figure 3 shows, the plots of $E$ form $K$ tight, well-separated clouds, confirming that the model has encoded each environment into a distinct region of the latent simplex. By contrast, the $U$ embeddings for WADI appear as scattered but well-separated clouds, whereas SMD forms coherent arcs. In both cases, coloring each $U$ point by $\arg\max(E)$ shows that every latent falls strictly within its corresponding $E$ cluster. This confirms that it always respects the discrete regimes encoded by $E$ even when $U$ is diffuse or varying. Overall, these results validate that (1) our encoder disentangles a small number of causal regimes in $E$, and (2) the primary latent $U$ varies within each regime, exactly as designed.

To testify that the model uses the environment code E at inference rather than treating it as a redundant head. We train the model normally with learned E. At test time we compare three settings: Default uses each window's inferred E; Single-E forces all windows to the most frequent E; Shuffled-E randomly permutes the inferred E across windows and recomputes the score. As Table 4 shows, default gives the best F1 on all

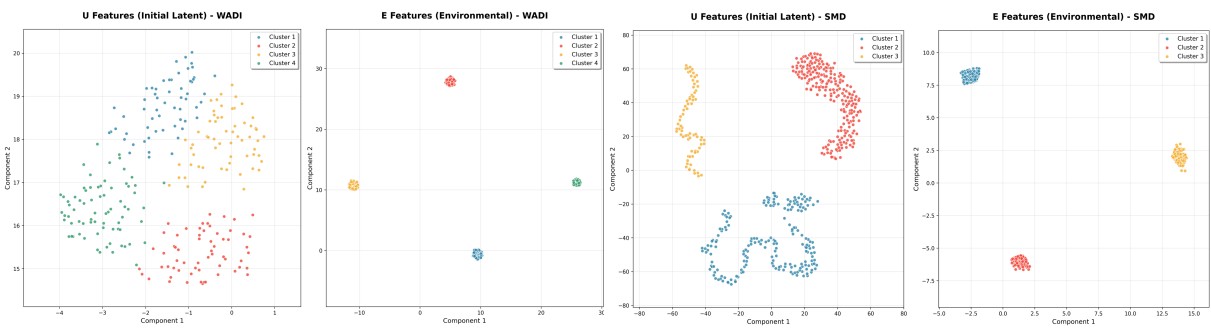

Figure 3: The two UMAP plots on the left show the embedded latents and environmental latents from the Causal Encoder for the WADI dataset. The two UMAP on the right present the corresponding latents for the SMD dataset.

Table 5: Effect of noise and missing-value settings on F1 performance (higher is better). Best scores for each dataset within a block are shown in **bold**. Default corresponds to training with no added noise or missing values. The Table 13 in Appendix is the extended version that includes the AUCPR results.

| Strategy | Level | ASD | ECG | MSL | SMAP | SMD | SWaT | WADI | PSM | Yahoo | KPI |
|---|---|---|---|---|---|---|---|---|---|---|---|
| | 0.10 | 0.652 | 0.584 | 0.501 | 0.475 | 0.569 | 0.761 | 0.701 | 0.680 | 0.889 | 0.218 |
| Noise level | 0.05 | 0.642 | 0.594 | 0.498 | 0.477 | 0.574 | **0.765** | **0.731** | 0.689 | 0.894 | 0.277 |
| | 0.01 | **0.703** | **0.600** | 0.497 | **0.480** | 0.575 | 0.762 | 0.720 | **0.690** | **0.978** | **0.353** |
| | 0.40 | 0.631 | 0.476 | 0.494 | 0.443 | 0.602 | 0.732 | 0.614 | 0.659 | 0.872 | 0.343 |
| Missing ratio | 0.20 | 0.605 | 0.476 | 0.495 | 0.465 | 0.599 | 0.754 | 0.652 | 0.675 | 0.872 | 0.343 |
| | 0.10 | 0.602 | 0.477 | 0.493 | 0.471 | **0.605** | 0.760 | 0.690 | 0.682 | 0.872 | 0.343 |
| Default | — | 0.676 | 0.584 | **0.529** | 0.478 | 0.575 | 0.763 | 0.716 | 0.686 | 0.966 | 0.346 |

datasets, Shuffled-E lowers F1 by about 0.13 on average, and Single-E is closer but still worse by 0.025 on average. These results show that correct environment assignments matter and that collapsing or misassigning E degrades performance.

## 4.4 Robustness under Data Perturbations

We evaluated the robustness of our model against two common data corruptions: additive **Gaussian noise** ($N(0, \sigma), \sigma \in 0.1, 0.05, 0.01$) and random **missing values** ($r \in 40\%, 20\%, 10\%$, imputed by mean). As Table 5 shows, mild noise often improved performance (e.g., ECG $0.584 \rightarrow 0.594/0.600$, WADI $0.716 \rightarrow 0.731/0.720$), with only minor drops at $\sigma = 0.1$ and scores still above baselines. Similarly, SMD benefited from masking ($0.575 \rightarrow 0.605/0.599/0.602$), while other datasets showed <10-point losses yet remained superior to competitors. These results highlight the framework's robustness under realistic perturbations.

## 4.5 Case Study: WADI

To better understand the semantics of the latent environment variable $E$, we conduct a qualitative case study from $\arg\max_k p(E_t = k \mid X_t)$ on the WADI dataset, where ground-truth attack intervals and predicted anomalies.

**Local attacks within a fixed environment** The left panel of Figure 4 focuses on the early portion of the trace that contains two kind of attacks, where motorised valve MV 001 is maliciously opened and flow transmitter 1 FIT 001 is turned off to induce incorrect chemical dosing. Because these only affects a single valve or sensor while leaving the overall network topology, control policy, and load pattern unchanged, these attacks act as local faults rather than a change of operating regime, so the latent environment $E$ is expected to remain the same. In both intervals the model keeps the environment constant at $E_1$, yet the anomaly score spikes and correctly flags the attacked windows as anomalous. This indicates that CSJD-AD does

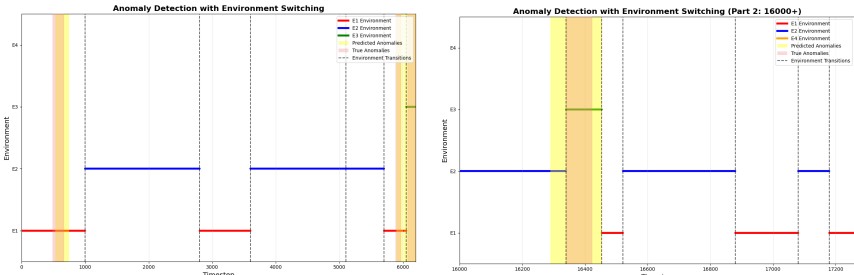

Figure 4: The plots show two sections of WADI datasets. The x-axis represents the timestep and the y-axis shows the latent environment status. The red shade is the regime-inconsistent deviations and the yellow shade is the predicted anomalies.

not explain away such local sensor faults by changing the environment; instead, they are interpreted as regime-violating deviations within an otherwise stable operating mode.

**Environment switches without anomalies** In the same panel we also observe several transitions between $E_1$ and $E_2$ (e.g., around time steps 1000 and 2800) that are not aligned with any documented attacks. These transitions correspond to slower, structurally driven changes in the underlying process (e.g., load or control adjustments) and are not accompanied by spikes in the anomaly alarm. The model therefore treats them as changes in the operating regime rather than anomalies, supporting our interpretation of $E$ as a coarse, environment-level context variable.

**Regime-shifting attacks that move both $E$ and the anomaly alarm** Figure 4 examines two system-level attacks involving the elevated reservoir: Around timestep $\approx 6100$ in the left panel, Attack 2 LIT 002 drains tank and manipulates water-quality readings, while Attack 15 is its inverse. Around time step $\approx 6100$ the onset of Attack coincides with a transition from $E_1$ to a distinct environment $E_3$. Notably, our detector already reacts before the environment switch is fully established: it is aware to early shifts in a subset of features and marks these early warning signs as anomalous. Similarly, near time step $\approx 16330$ Attack 15 (right panel) triggers a switch from $E_2$ to $E_3$, again alarmed by detector. In these cases the model correctly interprets the attacks as inducing new, degraded operating regimes, rather than as transient noise.

## 5 Conclusion

In conclusion, we propose an environment-conditioned soft jump-diffusion framework that unites continuous latent modeling with discrete environments, yielding state-of-the-art anomaly detection and enhanced interpretability. By pairing counterfactual and factual trajectories, our model, CSJD-AD quantifies regime, specific impacts and adapts to structural shifts via a jump-augmented SDE. This causal separation explains why an alarm is raised and delivers state-of-the-art detection performance across ten benchmarks. The approach thus offers a new, interpretable direction for TSAD by explicitly linking latent dynamics to regime-specific causal structure.

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

# A  Tech details

## A.1  Anonymous source code

Code is available at `https://anonymous.4open.science/r/CSJD-AD`.

## A.2  Training Resources

All experiments were carried out on a single desktop workstation with the following hardware and software configuration:

- **Operating System**: Ubuntu 24.04 LTS
- **CPU**: AMD Ryzen 9 9950X3D
- **System Memory**: 64 GB
- **GPU**: NVIDIA GeForce RTX 4090 (24 GB VRAM)
- **Libraries**: Python 3.8 + Pytorch 2.4.1 + CUDA 12.1

## A.3  Training Settings

To ensure consistency across experiments and to minimize the impact of individual hyper-parameter choices, we *fixed* the sliding-window length to 200 for *all* datasets—even though their respective optimal windows differ (details in Appendix C.1). Each model was trained for up to 200 epochs with early stopping, allowing adaptive convergence on each dataset.

Anomaly thresholds were selected via a grid search that maximized the $\ell_{\mathrm{F1}}$ score, thereby reducing sensitivity to threshold choice. During test, we denote the anomaly score at time $t$ as $\hat{S}_t$. The predicted label $\hat{y}_t \in \{0, 1\}$ is obtained by thresholding $\hat{S}_t$ with a fixed threshold $\delta$:

$$\hat{y}_t = \begin{cases} 1, & \text{if } \hat{S}_t > \delta, \\ 0, & \text{otherwise.} \end{cases} \tag{1}$$

Let $\mathbf{y} = \{y_1, \ldots, y_T\}$ be the ground-truth labels and $\hat{\mathbf{y}}(\delta) = \{\hat{y}_1, \ldots, \hat{y}_T\}$ the predicted labels induced by threshold $\delta$. We select the optimal threshold $\delta^*$ by

$$\delta^* \triangleq \arg\max_{\delta} \mathrm{F1}\big(\hat{\mathbf{y}}(\delta), \mathbf{y}\big). \tag{2}$$

Table 7 lists the hyper-parameters shared by every experiment; the remaining dataset-specific settings are given in Table 8.

We intentionally keep the number of tunable hyperparameters small. Unless otherwise stated, the core architecture and the objective weights are shared across datasets, and we fix the sliding-window length to 200 to reduce sensitivity to windowing choices. Dataset-specific settings (e.g., batch size, learning rate, and input/hidden dimension) are used primarily for computational feasibility and to accommodate dataset scale and dimensionality, rather than extensive per-dataset performance tuning. In practice, we recommend a simple default configuration for new deployments: use the shared architecture and loss weights, set K to a small value (e.g., 3–5) to capture coarse regimes, and keep the jump blending coefficient $\gamma$ fixed. We observe that performance is stable within this small range of K, consistent with the sensitivity analysis reported in Appendix D.

| ASD | https://github.com/zhhlee/InterFusion/tree/main/data |
|---|---|
| ECG | https://www.cs.ucr.edu/~eamonn/discords/ECG_data.zip |
| SMAP/MSL | https://www.kaggle.com/datasets/patrickfleith/nasa-anomaly-detection-dataset-smap-msl |
| SMD | https://github.com/NetManAIOps/OmniAnomaly/tree/master/ |
| SWaT/WADI | https://itrust.sutd.edu.sg/itrust-labs_datasets/dataset_info/ |
| PSM | https://github.com/kaist-dmlab/DualTF/tree/main/datasets |
| Yahoo | https://webscope.sandbox.yahoo.com/catalog.php?datatype=s&did=70 |
| KPI | https://github.com/NetManAIOps/KPI-Anomaly-Detection |

### A.4 Datasets Sources

### A.5 Training Resource Analysis

The training time and resources usage is listed in Table 6, which lists compare our models with standard VAE Kingma & Welling (2014) and Transformer Vaswani et al. (2017) framework, and latest TSAD models like Transformer-based Senstive-HUE Feng et al. (2024) and diffusion-based IGCL Zhao et al. (2025). The additional performance evaluation on the standard VAE Kingma & Welling (2014) and Transformer Vaswani et al. (2017) is included in Table 10.

In respect of training time, our model is fastest overall, about 36% faster than Transformer and 68% faster than IGCL. In most datasets like ASD, ECG, WADI, Yahoo and KPI our method is clearly faster, while on MSL and SMD, it is slightly slower than the average baseline, suggesting that for some mid-sized or specific characteristics, our architecture incurs a small time overhead. On the large univariate benchmarks Yahoo and KPI, our method achieves up to 8 times faster training than the baselines, indicating low per-sample overhead and good scalability to large time-series datasets

The VRAM usage is moderate, which is not as light as VAE and Transformer, but significantly lighter than IGCL and Sensitive-HUE. On challenging datasets like WADI, VRAM grows, but remains within a feasible range relative to other strong baselines.

Regarding computing resource usage in FLOPS, our model is substantially cheaper than IGCL and slightly cheaper than Transformer on average, at the cost of more FLOPS than the VAE

## B Jump Diffusion Proof

### B.1 Proof Sketch

Under the assumption that the environment process is piecewise constant and predictable, and that the drift, diffusion and the combined jump map satisfy global Lipschitz and linear-growth bounds Higham & Kloeden (2005), classical SDE theory guarantees a unique strong solution on each macro-interval. At each jump time, the Lipschitz jump map deterministically updates the state, preserving uniqueness across intervals. For simulation, we partition each interval of length $\Delta t$ into Euler–Maruyama Kloeden et al. (1995) micro-steps for the diffusion and apply the jump exactly; the only discretization error is $O(\Delta t)$ in mean square, yielding strong convergence of order $1/2$.

### B.2 Existence and Uniqueness

We assume the environment process $E_t$ is piecewise-constant and predictable ($E_t = E_{t_k}$ for $t \in [t_k, t_{k+1})$ and $E_{t_k}$ is $\mathcal{F}_{t_k^-}$-measurable). Impose global Lipschitz and linear-growth conditions on the diffusion coefficients and the jump map $G(u, e) = u + p_E(u, e)J_E(u, e)$: there exist $L, K > 0$ such that for all $u, v \in \mathbb{R}^d$ and each

Table 6: Comparison of training time, VRAM usage and Floating point operations per second (FLOPS) across datasets and models. All metrics are lower and better. The best score is in **bold** and the second-best is in underline

| Metrics | Model | ASD | ECG | MSL | SMAP | SMD | SWaT | WADI | PSM | Yahoo | KPI |
|---|---|---|---|---|---|---|---|---|---|---|---|
| Training time (min) | VAE | **11** | **6** | **15** | **24** | **33** | **36** | **39** | **27** | 69 | 123 |
| | Transformer | 17 | 9 | 22 | 34.5 | 47 | 52 | 57 | 39.5 | 98 | 176 |
| | Sensitive-HUE | 16 | 23 | 33 | 52.5 | 72 | 62 | 52 | 42.5 | 77 | **91** |
| | IGCL | 29 | 16 | 32 | 54.5 | 77 | 83.5 | 90 | 61 | 205 | 400 |
| | Ours | 15 | 10 | 28 | 43.5 | 59 | 54 | 49 | 38.5 | **14** | 99 |
| VRAM (GB) | VAE | 0.720 | 0.612 | 0.684 | 0.633 | **0.582** | **1.447** | **2.312** | **1.498** | 0.442 | **0.546** |
| | Transformer | 0.901 | 0.762 | 0.844 | 0.783 | 0.722 | 1.802 | 2.881 | 1.863 | 0.543 | 0.673 |
| | Sensitive-HUE | 0.840 | 0.952 | **0.550** | **0.586** | 0.621 | 5.577 | 10.533 | 5.542 | **0.441** | 0.979 |
| | IGCL | 1.233 | 0.786 | 1.023 | 0.923 | 0.823 | 4.863 | 8.902 | 4.963 | 0.640 | 0.823 |
| | Ours | **0.627** | **0.533** | 0.957 | 0.838 | 0.719 | 3.968 | 7.217 | 4.087 | 0.552 | 0.720 |
| FLOPS (million) | VAE | **9.28** | **6.33** | **89.33** | **55.34** | **21.34** | **175.57** | **329.79** | **209.56** | 32.07 | 10.74 |
| | Transformer | 13.26 | 9.01 | 127.62 | 79.06 | 30.49 | 250.81 | 471.12 | 299.37 | 45.81 | 15.35 |
| | Sensitive-HUE | 15.46 | 18.31 | 140.32 | 90.59 | 40.86 | 252.68 | 464.49 | 302.41 | 35.75 | 12.69 |
| | IGCL | 22.62 | 16.21 | 182.86 | 116.23 | 49.59 | 396.74 | 743.88 | 463.37 | 95.18 | 34.75 |
| | Ours | 11.72 | 10.10 | 160.21 | 99.17 | 38.12 | 221.64 | 405.15 | 282.68 | **6.54** | **8.63** |

Table 7: Shared hyper-parameter settings used in all experiments. $\gamma$: the scalar controlling the strength of jump influence; $\lambda_{\text{causal}}$: weight on causal loss; **LR**: learning rate

| Hyperparameter | Value |
|---|---|
| $\lambda_{\text{causal}}$ | 1 |
| $\lambda_{\text{KL}}$ | 0.01 |
| Sliding-window size | 200 |
| Training epochs | 200 |
| $\gamma$ | 0.8 |
| LR | $5e^{-4}$ |

environment $e$,

$$
\begin{aligned}
\|\mu_e(u) - \mu_e(v)\| + \|\sigma_e(u) - \sigma_e(v)\| &\le L\|u - v\|, \\
\|\mu_e(u)\|^2 + \|\sigma_e(u)\|^2 &\le K(1 + \|u\|^2), \\
\|G(u, e) - G(v, e)\| &\le L\|u - v\|, \\
\|G(u, e)\| &\le K(1 + \|u\|).
\end{aligned}
\tag{3}
$$

Spectral normalisation and weight clipping enforce these bounds in practice. Induction over $k$ then yields a unique strong solution: the diffusion part admits a unique solution on $(t_k, t_{k+1})$, and the Lipschitz jump map deterministically propagates the state to $U_{t_{k+1}}$, preserving uniqueness. Environment-driven variations are captured by $\mu_E$ and the soft-jump term $p_E J_E$; any residual deviation therefore signals a causal violation, aligning with our anomaly-detection objective.

### B.3 Numerical Approximation and Training

Each macro interval $\Delta t$ is subdivided into $N$ micro–steps of size $\delta t = \Delta t/N$. For $m = 0, \ldots, N - 1$ we perform the Euler–Maruyama update

$$
U_{k,m+1} = U_{k,m} + \mu_E(U_{k,m})\,\delta t + \sigma_E(U_{k,m})\sqrt{\delta t}\,\epsilon_{k,m},
$$

$$
\epsilon_{k,m} \sim \mathcal{N}(0, I),
\tag{4}
$$

starting with $U_{k,0} = U_{t_k}$. After the $N$ micro–steps we apply the instantaneous soft jump

$$
U_{t_{k+1}} = U_{k,N} + p_E(U_{k,N}, E)\, J_E(U_{k,N}, E).
\tag{5}
$$

Table 8: Dataset-specific hyper-parameter settings. **Input/Latent/Hidden Dim**: dimensionalities of input, latent state, and hidden layers; **K**: number of pre-specified environments

| Dataset | Input Dim | Latent Dim | Hidden Dim | $K$ | Batch Size |
|---------|-----------|------------|------------|-----|------------|
| ASD     | 19        | 32         | 64         | 4   | 32         |
| ECG     | 2         | 32         | 64         | 2   | 16         |
| MSL     | 55        | 128        | 256        | 4   | 32         |
| SMAP    | 25        | 64         | 128        | 4   | 32         |
| SMD     | 38        | 64         | 128        | 3   | 32         |
| SWaT    | 51        | 128        | 256        | 4   | 128        |
| WADI    | 127       | 256        | 512        | 4   | 256        |
| PSM     | 25        | 64         | 128        | 8   | 128        |
| Yahoo   | 1         | 32         | 64         | 4   | 16         |
| KPI     | 1         | 32         | 64         | 3   | 64         |

Table 9: F1 scores for different sliding-window sizes (higher is better). Best scores are in **bold**

| Window | ASD | ECG | MSL | SMAP | SMD | SWaT | WADI | PSM | Yahoo | KPI |
|--------|-----|-----|-----|------|-----|------|------|-----|-------|-----|
| 50  | 0.556 | 0.391 | 0.489 | 0.463 | 0.565 | 0.751 | 0.689 | 0.678 | 0.967 | 0.243 |
| 100 | 0.601 | 0.454 | 0.502 | 0.467 | 0.561 | 0.757 | 0.698 | 0.675 | **0.991** | 0.279 |
| 150 | 0.632 | 0.549 | 0.513 | 0.475 | 0.564 | 0.760 | **0.722** | 0.683 | 0.984 | 0.321 |
| 200 | 0.676 | **0.584** | 0.529 | **0.478** | 0.575 | **0.763** | 0.716 | **0.686** | 0.966 | 0.346 |
| 250 | **0.704** | 0.552 | **0.543** | 0.474 | **0.589** | 0.761 | 0.698 | 0.680 | 0.963 | **0.381** |

Because the jump is handled exactly, the only source of discretisation error lies in the diffusion part. Under the Lipschitz and growth conditions above:

$$\max_{k \leq K} \mathbb{E}\big[\|U(t_{k+1}) - U_{t_{k+1}}\|^2\big] \leq C\,T\,\Delta t, \tag{6}$$

where $T = t_K$ and $C$ depends on $L, K$ but not on $k$. Thus the scheme converges in mean square with order $1/2$ and provides stable gradients for end-to-end optimisation.

Thus, our piecewise diffusion with soft jump formulation inherits the expressive power of classical jump models, while, thanks to Lipschitz constraints and exact jump handling, retaining both differentiability and solid theoretical guarantees (existence, uniqueness, and numerical convergence).

### B.4 Justification of the one-jump-per-window approximation

We model the effect of jumps on a window $[t_k, t_{k+1}]$ by a single soft jump

$$U_F^{(k)} = U_{CF}^{(k)} + p_E^{(k)} J_E^{(k)}, \tag{7}$$

where $U_{CF}^{(k)}$ is the environment-conditioned diffusion outcome on this window, $p_E^{(k)} \in (0, 1)$ is a learned gate (regime-violation probability) and $J_E^{(k)}$ is the learned jump magnitude under environment $E$. We now show that this update is a first-moment–exact approximation of a classical compound–Poisson jump term on $[t_k, t_{k+1}]$.

**Lemma 1:** Let $\{N_t\}_{t \geq 0}$ be a Poisson process with intensity $\lambda_E$ and let $\{J_i\}_{i \geq 1}$ be i.i.d. jump marks with $\mathbb{E}[J_i \mid U_{t_k}, E] = m_E(U_{t_k}, E)$. Consider a jump–diffusion on the macro-interval $[t_k, t_{k+1}]$ of length $\Delta t = t_{k+1} - t_k$, whose cumulative jump contribution on this window is

$$\mathcal{J}_k = \int_{t_k}^{t_{k+1}} J\,dN_t = \sum_{i=1}^{N_k} J_i, \quad N_k := N_{t_{k+1}} - N_{t_k}. \tag{8}$$

Assuming $\lambda_E$ and the jump law are approximately constant on $[t_k, t_{k+1}]$, we have

$$\mathbb{E}[\mathcal{J}_k \mid U_{t_k}, E] = (\lambda_E \Delta t) \, m_E(U_{t_k}, E). \tag{9}$$

**Proof:** On a short interval $[t_k, t_{k+1}]$ with length $\Delta t$ and constant intensity $\lambda_E$, the increment $N_k$ is Poisson distributed, $N_k \sim \text{Poisson}(\lambda_E \Delta t)$, and is independent of the marks $\{J_i\}$. Hence

$$
\begin{aligned}
\mathbb{E}[\mathcal{J}_k \mid U_{t_k}, E] &= \mathbb{E}\Big[ \sum_{i=1}^{N_k} J_i \,\Big|\, U_{t_k}, E \Big] \\
&= \mathbb{E}[N_k] \, \mathbb{E}[J_i \mid U_{t_k}, E] \\
&= (\lambda_E \Delta t) \, m_E(U_{t_k}, E).
\end{aligned}
\tag{10}
$$

which proves the claim.

Lemma 1 suggests parameterising the expected jump effect on window $k$ by

$$p_E^{(k)} := \lambda_E \Delta t \in (0,1), \qquad J_E^{(k)} := m_E(U_{t_k}, E), \tag{11}$$

so that

$$\mathbb{E}[\mathcal{J}_k \mid U_{t_k}, E] = p_E^{(k)} J_E^{(k)}. \tag{12}$$

This is exactly the "one soft jump per window" update used in our model,

$$U_F^{(k)} = U_{CF}^{(k)} + p_E^{(k)} J_E^{(k)}, \tag{13}$$

which should therefore be understood as a first-moment approximation of a compound–Poisson jump component aggregated over $[t_k, t_{k+1}]$. We do not claim pathwise equivalence to a fully resolved jump process; instead, we match its expected contribution on each window, which is sufficient for our reconstruction and detection objectives.

## C    Hyperparameter Sensitivity Analysis

### C.1    Window Size Analysis

Table 9 presents the complete table of CSJD-AD performance across all datasets under varying window sizes (from 50 to 250). We observe that ASD, MSL, SMD, and KPI benefit from longer window sizes (250), while Yahoo, WADI, and ECG achieve better performance with shorter windows (100, 150 or 200). Based on these trends, we select a window size of 200 as a balanced configuration to minimize sensitivity to this hyperparameter across datasets.

### C.2    $\lambda_{\text{causal}}$ and $\lambda_{\text{KL}}$ analysis

We investigate the effect of $\lambda_{causal}$ and $\lambda_{kl}$ on model performance (F1 score) using two multivariate datasets (WADI, SMD) and one univariate dataset (KPI). The trends are shown in Figure 5. For $\lambda_{causal}$, we test values in $\{0, 0.2, 0.5, 1, 3\}$, and for $\lambda_{kl}$, we test values in $\{0, 0.002, 0.005, 0.01, 0.03\}$. Note that both coefficients include 0, which corresponds to removing the causal or KL loss term, respectively.

On the KPI dataset, removing either the causal or KL loss leads to the most severe performance degradation, indicating that the univariate setting is particularly sensitive to these loss components, whereas the multivariate datasets are less affected. Overall, the optimal values of $\lambda_{causal}$ and $\lambda_{kl}$ are 1 and 0.01, respectively.

### C.3    Number of Environment Analysis

In Figure 6, we visualize the F1 score trends for different numbers of environments on WADI, SMD, and KPI. We evaluate models with 0, 1, 2, 3, 4, and 6 environments, where 0 corresponds to removing the

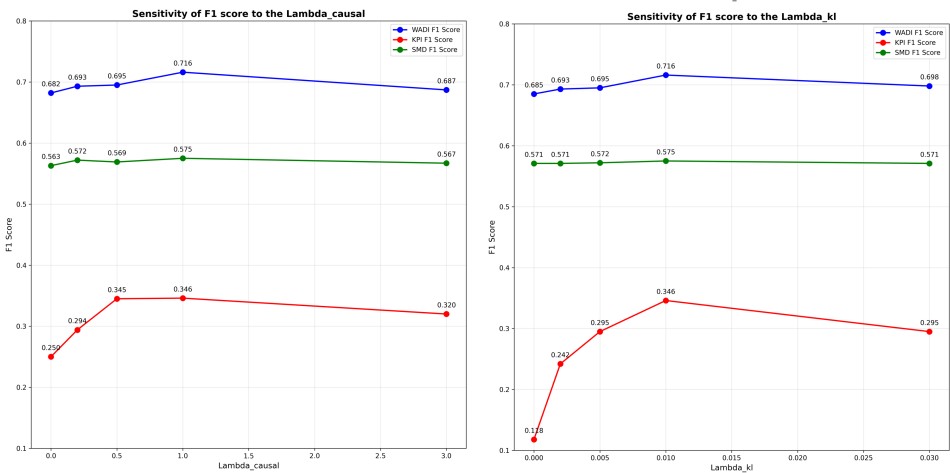

Figure 5: The left plot shows the F1 scores for different values of $\lambda_{causal}$, and the right plot shows the F1 scores for different values of $\lambda_{kl}$ on the WADI, SMD, and KPI datasets.

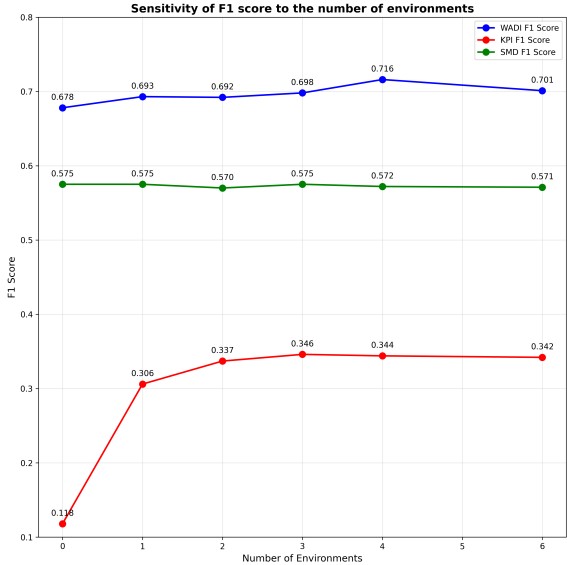

Figure 6: The visualization of F1 score of different number of environments on the WADI, SMD, and KPI datasets.

environment component entirely and using a standard encoder instead of a variational encoder to extract environment representations. KPI shows the highest sensitivity to the environment design, exhibiting a significant performance drop when the environment component is removed. In contrast, SMD is less sensitive and achieves nearly identical F1 scores when the number of environments is 0, 1, or 3. The optimal number of environments is 4 for WADI, 3 for SMD, and 4 for KPI.

### C.4 Gumbel-Softmax Temperature Analysis

The Gumbel–Softmax temperature controls the sharpness of the environment-selection gate. We evaluate temperatures (0.01, 0.05, 0.1, 0.2, 0.4) on WADI, SMD, and KPI (Figure 7). Across all three datasets, the F1 scores remain stable, showing only minor fluctuations as the temperature varies. A moderate temperature of 0.2 achieves the best overall performance, but the differences across the entire range are small. This

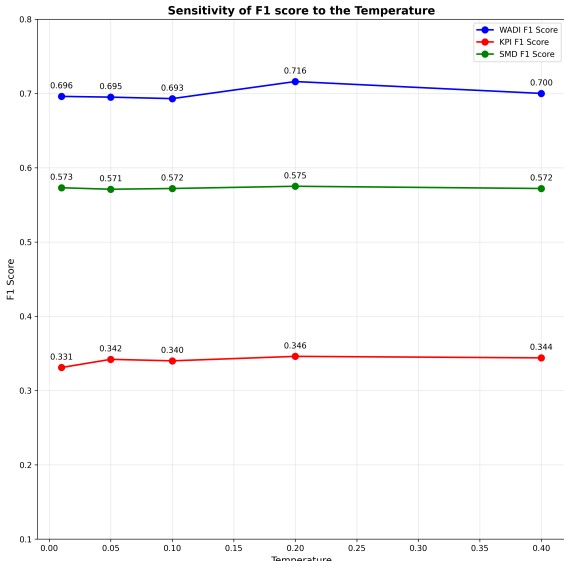

Figure 7: The visualization of F1 score of different temperature value selection on the WADI, SMD, and KPI datasets.

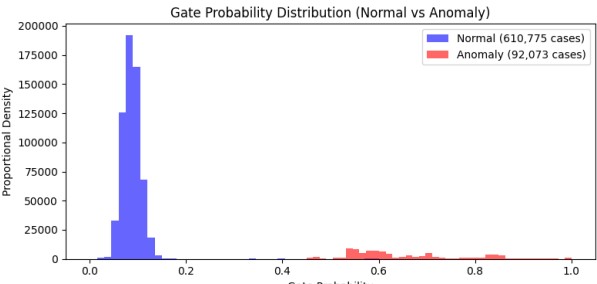

Figure 8: The visualization shows the distribution of gate probability values on the WADI dataset. The blue bars correspond to gate probabilities for normal data, while the red bars represent those for anomalous data.

confirms that CSJD-AD is not sensitive to the precise gating sharpness and remains robust under different temperature settings.

### C.5 Gate Probability Analysis

To better understand how the jump gate contributes to anomaly detection, we further examine the distribution of gate probabilities and the discriminative power of the gate alone. Figures 8 shows a clear separation between normal and anomalous windows on the WADI dataset: normal data exhibit very low gate probabilities, while anomalous windows concentrate near high gate values. This indicates that the learned gate responds strongly when the latent dynamics significantly violate the regime-specific diffusion pattern, rather than firing on benign fluctuations.

We also compute a gate-only ROC curve by treating the gate probability as an anomaly score (Figure 9). The resulting ROC-AUC of 0.8988 demonstrates that the gate itself is highly informative, even without using the full CSJD-AD detector. This behaviour confirms that the gating mechanism reliably captures regime-violating deviations and contributes meaningfully to overall sensitivity, consistent with our design of identifying anomalies as inconsistencies with environment-conditioned dynamics.

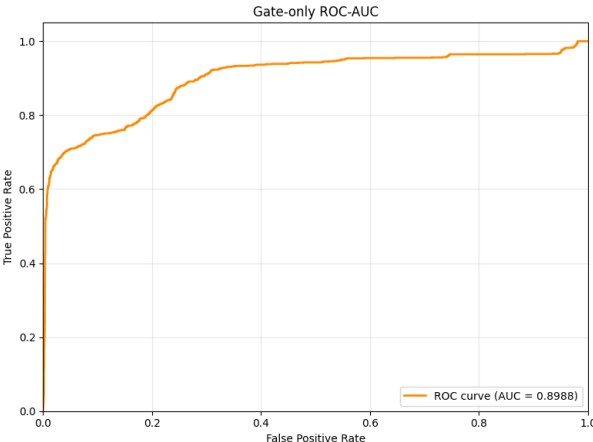

Figure 9: The ROC-AUC visualization of gate probability value and regime-inconsistent deviations.

## D Extended Benchmark Results

The Table 10 is the extended main results including precision and recall scores. The Table 11 and Table 13 is the extended ablation experiment and robustness test results with AUCPR additionally included.

## E Use of LLMs

We used ChatGPT to tidy and standardize LaTeX for mathematical formulas, harmonize notation, perform limited synonym substitutions to keep terminology consistent, and run light grammar checks on select sentences. We did not use LLMs to design the method, analyze data, select or curate results, write the experiments section, or generate synthetic data. We did not provide datasets, labels, or implementation code to the model. All technical content and claims were written and verified by the authors, and every LLM suggestion was reviewed and edited. The paper and results remain fully reproducible from our released code and data.

Table 10: Time-series anomaly-detection performance on ten public complete benchmarks with precision, recall, F1, and AUCPPR(higher is better). Best scores are in **bold**; second–best are underlined.

| Method | Metric | Multivariate Benchmarks | | | | | | | | Univariate Benchmarks | |
|---|---|---|---|---|---|---|---|---|---|---|---|
| | | ASD | ECG | MSL | SMAP | SMD | SWaT | WADI | PSM | Yahoo | KPI |
| VAE | Precision | 0.184 | 0.154 | 0.204 | 0.193 | 0.201 | 0.314 | 0.658 | 0.194 | 0.191 | 0.101 |
| | Recall | 0.417 | 0.329 | 0.646 | 0.231 | 0.464 | 0.184 | 0.115 | 0.342 | 0.360 | 0.217 |
| | F1 | 0.229 | 0.192 | 0.285 | 0.210 | 0.257 | 0.232 | 0.174 | 0.248 | 0.228 | 0.127 |
| | AUCPR | 0.171±.092 | 0.144±.072 | 0.199±.072 | 0.241±.194 | 0.276±.109 | 0.295 | 0.097 | 0.219 | 0.178±.071 | 0.095±.016 |
| Transformer | Precision | 0.301 | 0.232 | 0.324 | 0.301 | 0.297 | 0.421 | 0.803 | 0.491 | 0.182 | 0.112 |
| | Recall | 0.521 | 0.563 | 0.683 | 0.242 | 0.661 | 0.281 | 0.581 | 0.501 | 0.850 | 0.612 |
| | F1 | 0.382 | 0.329 | 0.440 | 0.268 | 0.410 | 0.337 | 0.674 | 0.496 | 0.300 | 0.189 |
| | AUCPR | 0.352±.102 | 0.421±.082 | 0.433±.091 | 0.317±.104 | 0.460±.142 | 0.291 | 0.635 | 0.481 | 0.521±.214 | 0.241±.051 |
| LSTM-VAE | Precision | 0.245 | 0.206 | 0.272 | 0.296 | 0.268 | 0.576 | 0.877 | 0.391 | 0.255 | 0.135 |
| | Recall | 0.521 | 0.411 | 0.808 | 0.830 | 0.580 | 0.970 | 0.144 | 0.574 | 0.450 | 0.271 |
| | F1 | 0.327 | 0.274 | 0.407 | 0.437 | 0.367 | 0.762 | 0.248 | 0.465 | 0.326 | 0.182 |
| | AUCPR | 0.245±.180 | 0.206±.150 | 0.285±.249 | 0.258±.305 | 0.395±.257 | 0.713 | 0.139 | 0.441 | 0.255±.152 | 0.135±.120 |
| OmniAnomaly | Precision | 0.167 | 0.147 | 0.161 | 0.196 | 0.306 | 0.658 | 0.994 | 0.421 | 0.219 | 0.133 |
| | Recall | 0.414 | 0.440 | 0.846 | 0.942 | 0.912 | 0.906 | 0.129 | 0.282 | 0.762 | 0.411 |
| | F1 | 0.238 | 0.216 | 0.271 | 0.325 | 0.459 | 0.762 | 0.229 | 0.338 | 0.340 | 0.201 |
| | AUCPR | 0.175±.132 | 0.154±.152 | 0.149±.182 | 0.115±.129 | 0.365±.202 | 0.713 | 0.120 | 0.391 | 0.245±.218 | 0.140±.010 |
| AnomalyTran | Precision | 0.298 | 0.325 | 0.218 | 0.266 | 0.206 | 0.971 | 0.057 | 0.491 | 0.260 | 0.212 |
| | Recall | 0.744 | 0.812 | 0.823 | 0.869 | 0.582 | 0.594 | 0.434 | 0.314 | 0.651 | 0.525 |
| | F1 | 0.425 | 0.464 | 0.344 | 0.407 | 0.304 | 0.737 | 0.102 | 0.403 | 0.372 | 0.303 |
| | AUCPR | 0.281±.201 | 0.306±.221 | 0.236±.237 | 0.264±.315 | 0.273±.232 | 0.681 | 0.040 | 0.471 | 0.261±.182 | 0.204±.139 |
| TranAD | Precision | 0.233 | 0.346 | 0.290 | 0.336 | 0.302 | 0.192 | 0.887 | 0.712 | 0.392 | 0.223 |
| | Recall | 0.446 | 0.691 | 0.759 | 0.788 | 0.534 | 0.796 | 0.155 | 0.472 | 0.630 | 0.401 |
| | F1 | 0.305 | 0.461 | 0.420 | 0.471 | 0.386 | 0.310 | 0.263 | 0.568 | 0.484 | 0.287 |
| | AUCPR | 0.238±.178 | 0.368±.251 | 0.278±.239 | 0.287±.300 | 0.412±.260 | 0.192 | 0.139 | 0.669 | 0.691±.324 | 0.285±.206 |
| $D^3R$ | Precision | 0.150 | 0.188 | 0.110 | 0.152 | 0.237 | 0.285 | 0.063 | 0.431 | 0.126 | 0.094 |
| | Recall | 0.751 | 0.751 | 0.930 | 0.381 | 0.526 | 0.195 | 0.831 | 0.391 | 0.501 | 0.375 |
| | F1 | 0.253 | 0.301 | 0.197 | 0.217 | 0.326 | 0.232 | 0.117 | 0.410 | 0.201 | 0.152 |
| | AUCPR | 0.150±.110 | 0.180±.131 | 0.138±.101 | 0.445±.218 | 0.228±.167 | 0.205 | 0.070 | 0.507 | 0.120±.080 | 0.090±.061 |
| PUAD | Precision | 0.263 | 0.285 | 0.258 | 0.194 | 0.269 | 0.362 | 0.955 | 0.331 | 0.225 | 0.211 |
| | Recall | 0.525 | 0.570 | 0.750 | 0.472 | 0.562 | 0.196 | 0.150 | 0.582 | 0.450 | 0.424 |
| | F1 | 0.351 | 0.382 | 0.384 | 0.275 | 0.364 | 0.254 | 0.259 | 0.422 | 0.301 | 0.284 |
| | AUCPR | 0.280±.203 | 0.304±.221 | 0.307±.102 | 0.319±.082 | 0.291±.210 | 0.271 | 0.155 | 0.449 | 0.240±.172 | 0.224±.152 |
| NPSR | Precision | 0.267 | 0.315 | 0.240 | 0.382 | 0.265 | 0.281 | 0.784 | 0.351 | 0.413 | 0.241 |
| | Recall | 0.525 | 0.788 | 0.839 | 0.104 | 0.623 | 0.582 | 0.500 | 0.584 | 0.825 | 0.483 |
| | F1 | 0.350 | 0.451 | 0.373 | 0.164 | 0.372 | 0.379 | 0.613 | 0.438 | 0.550 | 0.321 |
| | AUCPR | 0.281±.201 | 0.405±.281 | 0.336±.241 | 0.284±.142 | 0.335±.245 | 0.296 | 0.552 | 0.444 | 0.495±.344 | 0.288±.160 |
| DiffAD | Precision | 0.081 | 0.350 | 0.267 | 0.305 | 0.054 | 0.194 | 0.142 | 0.351 | 0.143 | 0.077 |
| | Recall | 0.402 | 0.143 | 0.025 | 0.291 | 0.025 | 0.105 | 0.500 | 0.491 | 0.765 | 0.764 |
| | F1 | 0.135 | 0.203 | 0.047 | 0.298 | 0.035 | 0.136 | 0.221 | 0.409 | 0.241 | 0.140 |
| | AUCPR | 0.523±.013 | 0.552±.025 | 0.321±.102 | 0.241±.083 | 0.102±.031 | 0.083 | 0.432 | 0.482 | 0.293±.124 | 0.231±.042 |
| Dual-TF | Precision | 0.620 | 0.480 | 0.116 | 0.119 | 0.263 | 0.143 | 0.504 | 0.461 | 0.665 | 0.303 |
| | Recall | 0.710 | 0.610 | 0.140 | 0.256 | 0.316 | 0.404 | 0.605 | 0.559 | 0.797 | 0.363 |
| | F1 | 0.661 | 0.538 | 0.127 | 0.163 | 0.287 | 0.212 | 0.551 | 0.506 | 0.725 | 0.330 |
| | AUCPR | 0.628±.212 | 0.511±.182 | 0.124±.126 | 0.141±.082 | 0.215±.074 | 0.171 | 0.523 | 0.354 | 0.689±.234 | 0.314±.107 |
| Sensitive-HUE | Precision | 0.286 | 0.215 | 0.330 | 0.178 | 0.295 | 0.972 | 0.865 | 0.402 | 0.167 | 0.099 |
| | Recall | 0.505 | 0.550 | 0.712 | 0.426 | 0.608 | 0.844 | 0.587 | 0.362 | 0.870 | 0.602 |
| | F1 | 0.366 | 0.309 | 0.451 | 0.251 | 0.397 | **0.904** | 0.699 | 0.381 | 0.281 | 0.170 |
| | AUCPR | 0.340±.188 | 0.410±.245 | 0.432±.121 | 0.319±.194 | 0.462±.283 | **0.873** | 0.641 | 0.681 | 0.489±.429 | 0.227±.253 |
| IGCL | Precision | 0.228 | 0.366 | 0.219 | 0.160 | 0.173 | 0.821 | 0.447 | 0.404 | 0.408 | 0.340 |
| | Recall | 0.011 | 0.054 | 0.228 | 0.297 | 0.259 | 0.638 | 0.007 | 0.464 | 0.133 | 0.150 |
| | F1 | 0.022 | 0.094 | 0.223 | 0.208 | 0.208 | 0.718 | 0.014 | 0.432 | 0.201 | 0.208 |
| | AUCPR | 0.079±.066 | 0.183±.141 | 0.179±.102 | 0.181±.061 | 0.126±.132 | 0.691 | 0.218 | 0.461 | 0.300±.277 | 0.206±.198 |
| RedLamp | Precision | 0.148 | 0.110 | 0.254 | 0.243 | 0.068 | 0.301 | 0.756 | 0.509 | 0.205 | 0.042 |
| | Recall | 0.336 | 0.334 | 0.321 | 0.153 | 0.328 | 0.101 | 0.532 | 0.022 | 0.548 | 0.087 |
| | F1 | 0.205 | 0.165 | 0.284 | 0.187 | 0.113 | 0.153 | 0.624 | 0.043 | 0.299 | 0.057 |
| | AUCPR | 0.154±.103 | 0.200±.196 | 0.199±.290 | 0.321±.306 | 0.128±.140 | 0.083 | 0.564 | 0.454 | 0.653±.409 | 0.089±.129 |
| Ours | Precision | 0.686 | 0.492 | 0.413 | 0.333 | 0.538 | 0.914 | 0.803 | 0.610 | 0.951 | 0.269 |
| | Recall | 0.666 | 0.719 | 0.733 | 0.846 | 0.617 | 0.655 | 0.646 | 0.782 | 0.980 | 0.486 |
| | F1 | **0.676** | **0.584** | **0.529** | **0.478** | **0.575** | 0.763 | **0.716** | **0.686** | **0.966** | **0.346** |
| | AUCPR | **0.682±.193** | **0.631±.176** | **0.464±.296** | **0.400±.317** | **0.637±.183** | 0.783 | 0.644 | **0.774** | **0.938±.200** | **0.342±.242** |

Table 11: Ablation study of the proposed model. Each column reports F1 scores (higher is better). **Bold** numbers denote the best result for that dataset.

| Metrics | Variant | ASD | ECG | MSL | SMAP | SMD | SWaT | WADI | PSM | Yahoo | KPI |
|---------|---------|-----|-----|-----|------|-----|------|------|-----|-------|-----|
| F1 | w/o $U_\text{F}$ | 0.562 | 0.573 | 0.523 | 0.443 | 0.563 | 0.751 | 0.691 | 0.672 | 0.892 | 0.194 |
| | w/o $p_E$ | 0.675 | **0.604** | 0.515 | 0.432 | 0.568 | 0.742 | 0.702 | 0.651 | 0.899 | 0.310 |
| | w/o $E$ | 0.571 | 0.552 | 0.505 | 0.403 | **0.575** | 0.733 | 0.706 | 0.673 | 0.811 | 0.118 |
| | w/o $\mathcal{L}_\text{causal}$ | **0.682** | 0.506 | 0.507 | 0.451 | 0.563 | 0.759 | 0.711 | 0.662 | 0.932 | 0.250 |
| | w/o $\mathcal{L}_\text{KL}$ | 0.680 | 0.578 | 0.512 | 0.471 | 0.571 | 0.741 | 0.710 | 0.679 | 0.934 | 0.188 |
| | Default | 0.676 | 0.584 | **0.529** | **0.478** | **0.575** | **0.763** | **0.716** | **0.686** | **0.966** | **0.346** |
| AUCPR | w/o $U_\text{F}$ | 0.559 | 0.615 | 0.461 | 0.382 | 0.621 | 0.771 | 0.633 | 0.767 | 0.881 | 0.210 |
| | w/o $p_E$ | **0.689** | **0.651** | 0.449 | 0.379 | 0.631 | 0.768 | **0.668** | 0.753 | 0.928 | 0.309 |
| | w/o $E$ | 0.569 | 0.596 | 0.454 | 0.352 | 0.636 | 0.749 | 0.629 | 0.765 | 0.821 | 0.180 |
| | w/o $\mathcal{L}_\text{causal}$ | 0.688 | 0.612 | 0.461 | 0.369 | 0.621 | 0.778 | 0.636 | 0.748 | **0.930** | 0.182 |
| | w/o $\mathcal{L}_\text{KL}$ | 0.686 | 0.620 | 0.462 | 0.392 | 0.632 | 0.757 | 0.632 | 0.771 | 0.872 | 0.247 |
| | Default | 0.682 | 0.631 | **0.464** | **0.400** | **0.637** | **0.783** | 0.664 | **0.774** | **0.938** | **0.342** |

Table 12: Effect of modifying learned environment settings on F1 and AUCPR performance (higher is better). Best scores for each dataset within a block are shown in **bold**. Default corresponds to training with correctly learned environment variables.

| Metrics | Strategy | ASD | ECG | MSL | SMAP | SMD | SWaT | WADI | PSM | Yahoo | KPI |
|---------|----------|-----|-----|-----|------|-----|------|------|-----|-------|-----|
| F1 | Single E | 0.654 | 0.581 | 0.512 | 0.450 | **0.575** | 0.759 | 0.709 | 0.663 | **0.966** | 0.306 |
| | Shuffled E | 0.539 | 0.467 | 0.504 | 0.408 | 0.396 | 0.732 | 0.698 | 0.651 | 0.811 | 0.143 |
| | Default | **0.676** | **0.584** | **0.529** | **0.478** | **0.575** | **0.763** | **0.716** | **0.686** | **0.966** | **0.346** |
| AUCPR | Single E | 0.572 | 0.626 | 0.450 | 0.357 | 0.641 | 0.731 | 0.647 | 0.752 | 0.929 | 0.341 |
| | Shuffled E | 0.670 | 0.503 | 0.409 | 0.291 | 0.438 | 0.672 | 0.642 | 0.748 | 0.537 | 0.139 |
| | Default | **0.682** | **0.631** | **0.467** | **0.400** | **0.637** | **0.783** | **0.664** | **0.774** | **0.938** | **0.342** |

Table 13: Effect of noise and missing-value settings on F1 and AUCPR performance (higher is better). Best scores for each dataset within a block are shown in **bold**. Default corresponds to training with no added noise or missing values.

| Metrics | Strategy | Level | ASD | ECG | MSL | SMAP | SMD | SWaT | WADI | PSM | Yahoo | KPI |
|---|---|---|---|---|---|---|---|---|---|---|---|---|
| F1 | Noise level | 0.10 | 0.652 | 0.584 | 0.501 | 0.475 | 0.569 | 0.761 | 0.701 | 0.680 | 0.889 | 0.218 |
| | | 0.05 | 0.642 | 0.594 | 0.498 | 0.477 | 0.574 | **0.765** | **0.731** | 0.689 | 0.894 | 0.277 |
| | | 0.01 | **0.703** | **0.600** | 0.497 | **0.480** | 0.575 | 0.762 | 0.720 | **0.690** | **0.978** | **0.353** |
| | Missing ratio | 0.40 | 0.631 | 0.476 | 0.494 | 0.443 | 0.602 | 0.732 | 0.614 | 0.659 | 0.872 | 0.343 |
| | | 0.20 | 0.605 | 0.476 | 0.495 | 0.465 | 0.599 | 0.754 | 0.652 | 0.675 | 0.872 | 0.343 |
| | | 0.10 | 0.602 | 0.477 | 0.493 | 0.471 | **0.605** | 0.760 | 0.690 | 0.682 | 0.872 | 0.343 |
| | Default | — | 0.676 | 0.584 | **0.529** | 0.478 | 0.575 | 0.763 | 0.716 | 0.686 | 0.966 | 0.346 |
| AUCPR | Noise level | 0.10 | 0.678 | 0.625 | 0.451 | 0.387 | 0.629 | 0.779 | 0.649 | 0.770 | 0.859 | 0.260 |
| | | 0.05 | 0.677 | 0.634 | 0.456 | 0.398 | 0.636 | **0.791** | **0.676** | 0.779 | 0.911 | 0.295 |
| | | 0.01 | **0.754** | **0.640** | 0.456 | **0.411** | 0.637 | 0.784 | 0.668 | **0.782** | 0.928 | **0.369** |
| | Missing ratio | 0.40 | 0.664 | 0.528 | 0.453 | 0.314 | 0.673 | 0.769 | 0.489 | 0.761 | 0.929 | 0.357 |
| | | 0.20 | 0.637 | 0.528 | 0.454 | 0.357 | 0.669 | 0.775 | 0.606 | 0.769 | 0.929 | 0.357 |
| | | 0.10 | 0.632 | 0.528 | 0.452 | 0.379 | **0.677** | 0.799 | 0.612 | 0.770 | 0.929 | 0.357 |
| | Default | — | 0.682 | 0.627 | **0.464** | 0.400 | 0.637 | 0.783 | 0.664 | 0.774 | **0.938** | 0.342 |

