# OpenReview forum: "A Causal Perspective on Soft Jump-Diffusion for Time-Series Anomaly Detection"
_TMLR — Under review for TMLR_

### Review · Reviewer_ZDG7 · 2026-06-27

**Summary Of Contributions:**

The paper claims that most TSAD methods assume a stationary data-generating process and overlook latent causal structures, and offers ideas to mitigate this.

**Additional Comments:**

None

**Audience:**

No

**Audience Explanation:**

I went to ECG https://www.cs.ucr.edu/~eamonn/discords/ECG_data.zip and downloaded the data. But there are no labels on the data. So..
1)	How did you obtain the labels? Yor report 3 significant digits, so you clearly believe you have very precise labels.
2)	How can the reviewers and readers obtain the labels, so we can reproduce the results?

You make a big deal about your success on ECG. “As shown in Table 2, our model demonstrates strong robustness under severe class imbalance. For instance, on the ECG datasets—both characterized by extremely low anomaly ratios—we achieve AUCPR scores0.627.” But when I look at the paper they came from (your “HOT SAX:” reference), it appears that discords did very well here 20 years ago, look at figures 1, 11, 12 , 13 ,14. That begs the questions, why not compare to time series discords, and if we could find these anomalies 22 years ago, does that mean they are too easy, or that we have made little progress in 22 years?

--

You test on YAHOO. But there are two problems with YAHOO.
1)	It is clear that the real YAHOO have lots of false positives and false negatives. Look at  A1Benchmark-real32 from 1240 to 1280. That is marked as normal, but it is clearly an anomaly.
2)	The YAHOO synthetic datasets are so simple and trivial that we can solve them with one line of code, or with 80 year old statistical process control algorithms. Lets see an example, A2Benchmark- synthetic1.  Lets plot it, with its label…
>> figure;, hold on; plot(synthetic1(:,2)/1000);,plot(synthetic1(:,3)-3,'r');,
It looks trivial to solve.  Lets solve it with a 80 year old SPC algorithm…
figure;, hold on; plot(diff(synthetic1(:,2))> std(synthetic1(:,2)), 'g');,plot(synthetic1(:,3)-3,'r');,

To summarize, I don’t think we can be impressed with any results on YAHOO, it is embarrassingly trivial and the amount of mislabeling means we could not trust the differences between algorithms in any case.

--
You test on PSM, In this dataset 27.76% of the data is anomalous!!!  This alone would worry me. The normal  definition of anomaly  is “deviation from the normal or usual order, type”, how can you call these things anomalies if they are one third of the data?

Moreover, at least some of these anomalies are trivial

1)There is an anomaly marked at 74450, but look at column 23. There is a spike there that is an order of magnitude larger than the surrounding data.
2) There is an anomaly marked at 32510, but look at column 23. There is a spike there that is two orders of magnitude larger than the surrounding data.
3)	There is a cluster of three anomalies marked at 10440, 10460 and 10520, but look at column 22. There is a spike there that is four orders of magnitude larger than the surrounding data.
4)	Look at the anomaly region from 7063 to 7398, now look at column 18 in the same region, how could ANY method fail to find these.
This dataset is embarrassingly trivial. We don’t need complex algorithms here.
---
You test on MSL and report 3 significant digits on it, which implies that you think the ground truth labels are accurate to at least one part in a thousand. Is that true? Lets take a look.
In particular let's look at G-1. The only anomaly labeled in 4770 to 4890. However, 4270 to 4285 and 6880 to 6894 are anomalies too. If a good algorithm finds these two extra anomalies, you will unfairly penalize that algorithm.

--
The ASD and KPI (and other datasets from Dan Pei’ lab are beyond a joke)
--
SWaT is nonsense for TSAD. “…Thus we conclude that evaluations on SWaT are highly unreliable and that these datasets are not suited for multivariate time-series AD evaluation. “ [v]. Look at SwaT-FIT401 from 227900 to 263700. Should we take credit for discovering an anomaly that a multiple orders of magnitude change in mean?


[v] Dennis Wagner, Tobias Michels, Florian C. F. Schulz, Arjun Nair, Maja Rudolph, Marius Kloft: TimeSeAD: Benchmarking Deep Multivariate Time-Series Anomaly Detection. Trans. Mach. Learn. Res. 2023 (2023) ml.informatik.uni-kl.de/publications/2023/TimeSeAD.pdf
[c] R Wu,Current Time Series Anomaly Detection Benchmarks are Flawed and are Creating the Illusion of Progress. IEEE Trans. Knowl. Data Eng. 35(3): 2421-2429 (2023) https://arxiv.org/abs/2009.13807

**Broader Impact Concerns:**

None.

**Claims And Evidence:**

No

**Claims Explanation:**

The paper only evaluates on datasets which are known to be trivial to solve and/or have errors in their ground truth that exceeds the differences between algorithms that is claimed. As such, we cannot evaluate the contribution [c].

**Requested Changes:**

Would it be possible to test on datasets that are non-trivial, and have better ground truth labels?

---

### Review · Reviewer_2j8Z · 2026-07-05

**Summary Of Contributions:**

This paper tackles the problem of anomaly detection for time series, which is particularly useful when regime-dependent mechanisms and structural shifts govern what we consider "normal" temporal dynamics. To address this problem, the paper introduces CSJD-AD, a framework that leverages context-conditioned jump diffusion processes, causality, and counterfactuals to force the model to learn which changes are expected under which regime. The method is evaluated on several datasets and benchmarks, achieving state-of-the-art performance and demonstrating the usefulness of the combined components for time-series anomaly detection.

This work has many strenghs, but is also greatly diminished by significant weaknesses.




### Key Strengths

1) A very strong introduction, featuring compelling examples to ground the work being done.
2) A very important improvement over traditional jump-diffusion models (Merton, 1976) by using soft gating instead of hard binary jumps.
3) Good theory and proofs (which I did not verify step-by-step) to back up the methodology. The evaluation metrics are also excellent, and the experimentation across 11 time-series anomaly detection baselines is extensive.

### Key Weaknesses

1) The paper introduces many components (see Contributions 1 to 4) and loss terms in Equations 8 to 10. While each is justified and the combination works very well, it is not clear how each individual component contributes to the success of the framework.

2) While the literature review is extensive, the authors only cover how CSJD-AD differs from existing diffusion-based methods. The authors should clarify how the framework builds upon pattern-deviation-based, forecasting-based, and reconstruction-based methods as well.

3) The presentation requires significant improvement. The authors should separate implementation challenges from core methodological concepts. This clarification is particularly necessary because Figure 2, dedicated to explaining the overall approach, is lacking (its caption describes $U$, $E$, and other terms not present in the diagram). To be further precise:
	- a)Some notation is peculiarly inconsistent, such as introducing a subscript $s$ in Section 3.4, but using a superscript in the ensuing Equation 5.
	- b) Before Equation 6, factuals and counterfactuals are not defined, yet the subscript $CF$ is still used in the equation. This notation overload obscures the methodology.

**Additional Comments:**

None

**Audience:**

Yes

**Audience Explanation:**

The general sequence modelling and diffusion communities, at the very elast, stand to benefit from this work.

**Claims And Evidence:**

Yes

**Claims Explanation:**

The claims made are supported by empirical evidence. The submission avoids making bold statements, as every major methodological claim is supported by corresponding quantitative experiments, including qualitative case studies.

**Requested Changes:**

### Critical for Securing Recommendation

1. The authors need to provide stronger ablation studies. While they conducted some, the results are reported without standard deviations, error bars, or confidence intervals. This is problematic, especially given that some ablations lead to closely comparable results (e.g., w/o $E$ on SMD in Table 3), or even outperform the default CSJD-AD.

2. The authors must provide a more comprehensive review of existing methods and explain how their contribution differs (see Weakness 2).

3. Finally, the authors should improve their presentation significantly, ranging from the mathematical notation to the figures. I provided examples of significant presentation issues in Weakness 3, but more are present throughout the paper (see below).


### Optional, but Would Strengthen the Work

1. Textual citations are incorrectly used in place of parenthetical citations, which disrupts the grammatical flow of the text. For example, at the start of page 2, it reads "...underlying causal mechanisms Carvalho et al. (2023)" but should be formatted as "...underlying causal mechanisms (Carvalho et al., 2023)." This might be due to the use of `\citet` instead of `\citep`? Please review the entire submission and address this.

2. There is a grammatical error: "These context-dependent variations are not anomalies by itself" should be corrected to "themselves".

3. Could the authors clarify the difference between an environment-consistent change and a regime-consistent one? (The terms "environment" and "regime" are used frequently throughout the paper without clear distinction.)

---

### Review · Reviewer_uq8U · 2026-07-13

**Summary Of Contributions:**

Summary: This paper proposes CSJD-AD, a time-series anomaly detection framework that models latent regime-dependent dynamics through an environment-conditioned soft jump-diffusion process. It infers discrete operating environments, generates counterfactual no-jump and factual jump-augmented trajectories, and scores anomalies as unexplained structural deviations. By combining reconstruction, variational regularization, and causal contrastive learning, the method aims to distinguish benign regime shifts from true anomalies and reports strong results across ten benchmark datasets.

Strength:
 - The paper addresses a meaningful limitation of existing TSAD methods: treating regime changes as anomalies under stationary assumptions. Its environment-conditioned formulation is conceptually well motivated and directly targets the practical need to distinguish normal operating shifts from unexpected faults.
 - The proposed factual–counterfactual trajectory design provides a clear modeling intuition. By comparing diffusion-only evolution with jump-augmented evolution, the method offers a structured way to separate expected regime-consistent dynamics from potentially anomalous deviations.
 - The WADI case study improves interpretability by showing how the model handles local attacks, benign environment switches, and regime-shifting attacks. This qualitative analysis helps connect the latent environment variable to concrete anomaly-detection behavior.

Weakness:
 - The main architecture figure does not clearly explain the functional role of each component, especially the distinction between environment inference, counterfactual path generation, factual jump injection, and anomaly scoring. A clearer pipeline diagram with step-by-step annotations would make the method much easier to understand.
 - Although the paper compares against several representative TSAD methods, many baselines are relatively old, such as LSTM-VAE, OmniAnomaly, TranAD, and DiffAD. The comparison includes only a limited number of very recent methods from the past one or two years, which weakens the claim of state-of-the-art performance.
 - The discrete environment variable EEE is central to the method, but the paper treats it as a semantic-free mechanism index. While this is practical for anonymized benchmarks, it also limits the interpretability claim, since the learned regimes are not clearly mapped to physical operating states.
 - The method involves variational encoding, environment-conditioned SDE dynamics, counterfactual/factual path generation, and per-entity training in some benchmarks. Although the paper reports resource usage, it would benefit from a clearer discussion of scalability in real-time or large-scale industrial monitoring settings.

**Audience:**

Yes

**Audience Explanation:**

The paper should be of interest to researchers working on time-series anomaly detection, neural stochastic differential equations, regime-aware representation learning, and interpretable anomaly detection. The proposed distinction between environment-consistent changes and jump-like anomalous deviations is methodologically relevant and practically meaningful.

**Broader Impact Concerns:**

There are not concerns that that would require adding a Broader Impact Statement.

**Claims And Evidence:**

Yes

**Claims Explanation:**

The experimental results generally support the main claims that CSJD-AD is an effective regime-aware time-series anomaly detection method. The ablation studies, environment intervention experiments, robustness tests, and WADI case study provide useful evidence for the proposed components, though some causal and interpretability claims could still be stated more carefully.

**Requested Changes:**

- Improve the clarity and visual quality of the architecture figures.
 - Add more recent TSAD baselines.
 - Clarify the causal interpretation of the proposed model.
 - Better explain the semantic meaning and limitation of the environment variable E.
 - Discuss scalability for real-time and large-scale monitoring.